# Multiple configurations and fluctuating trophic control in the Barents Sea food-web

Elliot Sivel[1,2]*, Benjamin Planque[1], Ulf Lindstrøm[1,2], Nigel G. Yoccoz[2]

**1** Institute of Marine Research, Ecosystem Processes Group, Fram Centre, Tromsø, Norway, **2** Department of Arctic and Marine Biology, UiT The Arctic University of Norway, Tromsø, Norway

* Elliot.sivel@hi.no

## Abstract

The Barents Sea is a subarctic shelf sea which has experienced major changes during the past decades. From ecological time-series, three different food-web configurations, reflecting successive shifts of dominance of pelagic fish, demersal fish, and zooplankton, as well as varying trophic control have been identified in the last decades. This covers a relatively short time-period as available ecological time-series are often relatively short. As we lack information for prior time-periods, we use a chance and necessity model to investigate if there are other possible configurations of the Barents Sea food-web than those observed in the ecological time-series, and if this food-web is characterized by a persistent trophic control. We perform food-web simulations using the Non-Deterministic Network Dynamic model (NDND) for the Barents Sea, identify food-web configurations and compare those to historical reconstructions of food-web dynamics. Biomass configurations fall into four major types and three trophic pathways. Reconstructed data match one of the major biomass configurations but is characterized by a different trophic pathway than most of the simulated configurations. The simulated biomass displays fluctuations between bottom-up and top-down trophic control over time rather than persistent trophic control. Our results show that the configurations we have reconstructed are strongly overlapping with our simulated configurations, though they represent only a subset of the possible configurations of the Barents Sea food-web.

## Introduction

Marine ecosystems have complex structures, functions, and dynamics. They include a diversity of species, comprise various habitats, and often display non-linear feedback mechanisms [1–3]. Interactions emerging at higher levels of the system can result from interactions and processes occurring at lower levels and vice versa [1]. Marine ecosystems are dynamic and constantly undergo structural and functional changes [4–7], evolving through different states or "configurations". A number of ecological studies focus on how external pressures may explain the differences between configurations of a same ecosystem at different time-periods [5,7,8]. However, changes in ecosystem configurations can also result from the internal dynamics emerging from its functioning or from external, stochastic events [9].

**Data Availability Statement:** The data is available in the manuscript and its Supporting information files.

**Funding:** This work was supported by the Norwegian Research Council project 276730 The

Nansen Legacy. The Norwegian Research Council had no role in the study design, data collection and analysis, decision to publish, or preparation of the manuscript.

**Competing interests:** The authors have declared that no competing interests exist.

The historical range of variability informs us on how an ecosystem varied in the past. It can be used as a reference for the variability of an ecosystem to improve the assessment and the prediction of future changes in its dynamics [10]. Available ecological time-series used to estimate the historical range of variability of marine ecosystems are often relatively short, typically less than 50 years [11,12]. Furthermore, ecological time-series are not available for all trophic groups (e.g. benthos, marine mammals, birds) and the lifespan of some species exceeds 50 years. Consequently, it is difficult to make inference about the past and future variability of marine ecosystems without making assumptions that can result in large uncertainties [11,13]. Yet, it is expected that ecosystem variability increases with the length of the study period [14,15], and that the baselines used for the assessment of fish abundances have shifted since 1950 [16,17]. To better anticipate possible changes in future marine ecosystem dynamics, one must explore the variability of the system dynamics on a timescale longer than the 50 years for which data are available. In some instances, these dynamics can be reconstructed using paleontological or archeological data (e.g. [18]). An alternative approach is to use numerical models to simulate dynamics over multidecadal time-periods.

Numerical food-web models, which represent one specific type of ecosystem models, integrate available data to simulate the dynamics of food-webs [19]. These models focus on trophic interactions between a subset of species or functional groups in the ecosystem. In the search of more realism, food-web models have become more complex by adding more parameters and variables for which data are not always available [20] thus increasing both the structural and parameter uncertainties. Despite this increase in complexity, these models are too constrained and are not able to reproduce the observed variability patterns in the ecosystem [6,21].

An alternative modelling approach, proposed by Mullon and collaborators [22], addresses specifically the issue of ecosystem stochastic variability. The Non-Deterministic Network Dynamic model (NDND) is a mass-balanced stochastic food-web model based on the principles of chance and necessity [23]. Chance implies that any event is possible as long as it complies with a number of basic biological and physical constraints ensuring that the ecosystem sustains itself over time. As such, the NDND model has been designed to reproduce the high variability of natural systems by exploring their "state-space". It thereby allows us to explore many possible food-web configurations (i.e. distribution of biomass among the species, trophic controls between the compartments) and the temporal variability in these configurations. A more recent study [24] showed that the NDND model was able to reproduce the variability patterns of several ecosystem properties (e.g. stability, trophic control, density dependence, etc.) observed in the Barents Sea food-web during the last three decades.

The Barents Sea is a subarctic shelf sea which has experienced major changes in species composition and dynamics during the past decades [7,8,25]. Available ecological time-series of the Barents Sea species revealed successive changes in food-web configurations reflecting shifts of dominance between pelagic fish, demersal fish [8], and zooplankton [25]. A past modelling study identified two distinct trophic pathways in the Barents Sea food-web based on an Ecopath model [26]: one pelagic and one benthic. Yaragina and Dolgov [27] suggested a wasp-waist control of the Barents Sea food-web where pelagic fish is assumed to exert a top-down control on zooplankton and a bottom-up control on its predators, whereas, Johannesen and collaborators [8] identified fluctuating trophic control between the species of the Barents Sea food-web. However, there is a need to define a baseline for the variability of the Barents Sea ecosystem to assess if the recent changes in its dynamics may reflect its stochastic variability or if they were induced by variations in anthropogenic drivers.

In this study, we investigate the variability of the Barents Sea food-web configurations emerging from random trophic interactions and how they differ from the configurations observed during the past three decades. To this end, we perform food-web simulations using

the NDND model for the Barents Sea food-web and analyze the biomass configurations and trophic pathways of these simulated food-webs and compare past vs. simulated food-web configurations. Furthermore, we investigate if the previously reported trophic control (top-down or bottom-up) of the Barents Sea food-web are persistent features of the Barents Sea food-web.

## Material and methods

### A. NDND model parametrization

The NDND model is a stochastic mass-balanced food-web model in which the food-web topology (i.e. who eats who) is fixed but the predation rates are indeterminate [22–24]. In the NDND model, the dynamics of the different trophospecies (hereafter named species) result from biomass exchanges between species whose values are sampled randomly (chance), given a set of biological and physical constraints (necessity). Trophic flows define mechanistically the biomass at the next time-step according to the master equation of the model (see supplementary materials S1 in S1 File). Estimated biomass values will then constrain the values of the trophic flows for the next time-step, and so on. Five constraints are defined for the NDND model: (1) the food intake of a predator is limited due to satiation, (2) the increase of biomass per time-step is limited, as well as (3) the decrease of biomass per time-step, (4) a trophic flow between two species cannot be negative, and (5) the biomass of species must stay above a threshold value referred to as the refuge biomass. The mathematical formulation of the NDND model is detailed in supplementary material S1 in S1 File.

In the present study, we used the food-web topology from Lindstrøm *et al*. [24] comprising eight species (phytoplankton, herbivorous zooplankton, omnivorous zooplankton, benthos, pelagic fish, demersal fish, mammals, and birds) and 18 trophic links (Fig 1). The values of the inertia parameter were adjusted to account for asynchronous growth rates of different species within species groups. Parameter values are presented in Table 1.

Harvesting of pelagic and demersal fish is explicitly included. In contrast to Lindstrøm *et al*. [24], who modelled fish catches as a constant fraction of biomass, we express fishing using a harvest control rule (HCR) that resemble the fishing regulations that operate in the Barents Sea [28]. To apply the HCR to the NDND model, we make the following assumptions: (1) we assume the ratio catch/stock to be equivalent in biomass and numbers, (2) the fishing mortality is estimated based on the total stock biomass instead of the spawning stock biomass, and (3) as capelin and cod are the major species of the pelagic and demersal fish groups, we use the parameter values from these two species to construct the HCR for the pelagic and demersal fish groups respectively.

The HCR is defined as follows:

1. when the stock biomass is smaller than the trigger biomass (*Blim*), then the catch is set to 0,

2. when the stock biomass is greater than *Blim* and lower than the target biomass (*Bmp*), the catch increases linearly with the total stock biomass,

3. when the stock biomass is larger than *Bmp* then, the caught biomass is estimated using the Baranov equation [29,30].

Parameter values of *Fmp*, *Bmp* and *Blim* are set according to [28] and the natural mortality rates (*M*) are set according to [31]. Parameter values of the HCR are presented in Table 2.

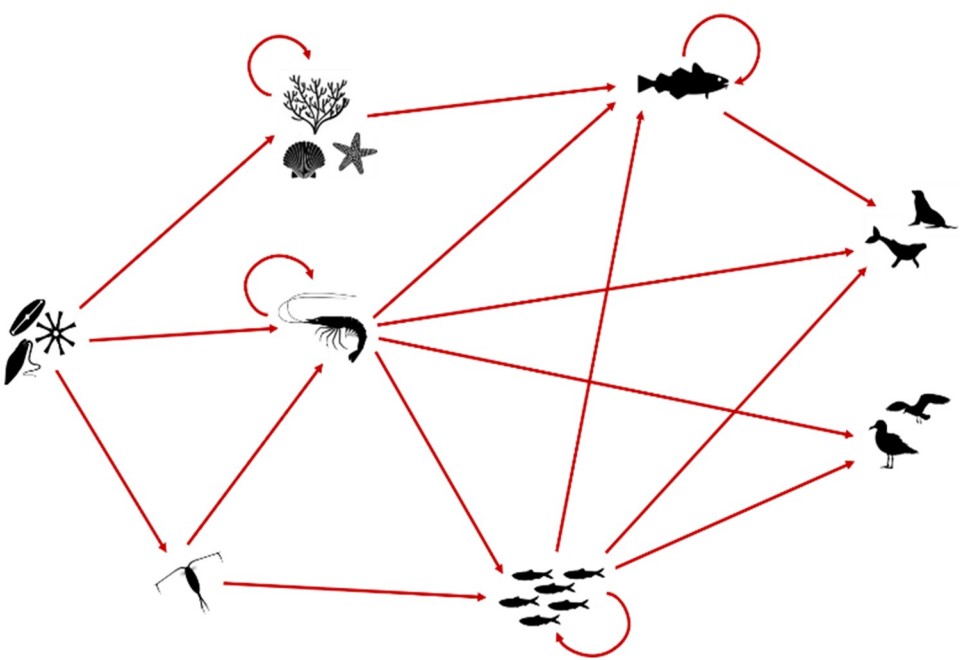

**Fig 1. Schematic of the simplified Barents Sea food-web.** Species are represented by icons, from the left to the right: Phytoplankton, herbivorous and omnivorous zooplankton, benthos, pelagic and demersal fish, mammals, and birds. Trophic links are represented by red arrows. Circular arrows correspond to cannibalistic interactions.

**Table 1. Parameter input for the simulation with the NDND model.**

|  | Phytoplankton | Herbivorous zooplankton | Omnivorous zooplankton | Benthos | Pelagics | Demersals | Mammals | Birds |
|---|---|---|---|---|---|---|---|---|
| Initial Biomass ($B_0$, tons·km$^{-2}$) | 25 | 23 | 12.9 | 66 | 0.36 | 1.18 | 0.34 | 0.007 |
| Import ($I$, tons·km$^{-2}$) | 1000 | 8 | 2 | 0 | 0 | 0 | 0 | 0 |
| Export ($E$, tons·km$^{-2}$) | 0 | 0 | 0 | 0 | 0 | 0 | 0 | 0 |
| Assimilation efficiency ($\gamma$) | 1 | 1 | 1 | 0.94 | 0.9 | 0.93 | 1 | 0.84 |
| Digestibility ($\kappa$) | 0.65 | 0.9 | 0.9 | 0.6 | 0.9 | 0.85 | - | - |
| Other losses ($\mu$) | 6.74 | 8.4 | 5.5 | 1.5 | 2.85 | 1.65 | 5.5 | 60 |
| Inertia ($\rho$) | 12.94 | 7.58 | 3.1 | 0.74 | 0.9 | 0.25 | 0.11 | 0.81 |
| Satiation ($\sigma$) | - | 128 | 42 | 25.2 | 13.5 | 5.5 | 10.9 | 123 |
| Refuge biomass ($\beta$, tons·km$^{-2}$) | 0.25 | 0.23 | 0.13 | 0.66 | 0.025 | 0.023 | 0.0034 | 0.0001 |

Assimilation efficiency and digestibility are ratio and do not have units. Other losses, inertia and satiation do not have units but represent ratios over a 1-year time-period.

**Table 2. Harvesting control rule input parameter values.**

|  | Phytoplankton | Herbivorous zooplankton | Omnivorous zooplankton | Benthos | Pelagics | Demersals | Mammals | Birds |
|---|---|---|---|---|---|---|---|---|
| Fishing mortality rate ($Fmp$) | - | - | - | - | 0.05 | 0.4 | - | - |
| Target biomass ($Bmp$, tons·km$^{-2}$) | - | - | - | - | 0.125 | 0.475 | - | - |
| Trigger biomass ($Blim$, tons·km$^{-2}$) | - | - | - | - | 0.125 | 0.25 | - | - |
| Natural mortality ($M$) | - | - | - | - | 0.85 | 0.2 |  |  |

## B. Simulations

All simulations and statistical analysis are performed using R (v.3.6.2) [32]. A summary of all data transformations and statistical analysis methods used in this study is given in supplementary material S2 in S1 File.

The sampling algorithm used in this study is the Complex Polytope Gibbs Sampling algorithm (cpgs) available in the RCaN package [33].

To ensure an extensive exploration of the "state-space" of the system, we generated 1000 trajectories of 500 years each. Possible configurations of the food-web can be defined based on the distribution of biomass in the different trophospecies or based on the distribution of trophic flows. To ensure that the simulated biomasses used for statistical analysis were independent from the initial biomasses at the start of the simulations, we removed a burn-in period of 121 years, leaving us with 1000 time-series of 379 years (biomass) and 378 years (flows) each. The method used to estimate the burn-in period is described in supplementary material S3 in S1 File. Finally, biomass data were log10 transformed to remove the scale difference between species. For the same purpose but, as some flows are close to zero, the flow data were log10(x + 0.001) transformed.

In the NDND model, phytoplankton biomass time-series reflect the remaining primary production after consumption by predator rather than the actual standing stock of algae or bacteria. These time-series were therefore not included in the analysis.

In addition to the NDND simulations, we used reconstructed trajectories of the Barents Sea food-web for the period 1988–2019 estimated from a Chance and Necessity model (CaN) for the Barents Sea [34]. In essence, NDND and CaN trajectories result from the same dynamical process but while the NDND trajectories are only constrained by ecological and biological limits, the CaN trajectories are also constrained by historical time-series of abundance and food consumption. Fishing in the CaN model is reconstructed as a non-trophic flow that is constrained by historical landing time-series for omnivorous zooplankton, pelagic fish, demersal fish, benthos, and marine mammals. The CaN model outputs provide an ensemble of biomass and trophic flow reconstructions for the whole food-web using available data series from a subset of species groups only. CaN model outputs are hereafter referred to as "reconstructed data". The details of the CaN model setup are provided in Supplementary material S4 in S1 File.

## C. Statistical analysis

A main goal of this study is to explore possible food-web configurations in the Barents Sea food-web and compare those to previously observed configurations. We define food-web configurations as either 1) the relative distribution of biomass among species in the food-web or 2) the relative distribution of trophic flows. Because the configurations are multivariate, a space reduction method such as principal component analysis (PCA) is useful to summarize the configurations.

Principal component analysis (PCA) has been widely used to identify ecosystem configurations [8,25,35]. PCA is best suited for cross-sectional studies, but as pointed out by Planque and Arneberg [36], the patterns emerging from PCA performed on multivariate autocorrelated time-series can be spurious. In such situations, an alternative statistical approach should be considered. Dynamical principal component analysis (dPCA) is a statistical method aiming at performing a PCA accounting for temporal autocorrelation [37,38]. In dPCA, time-lagged data are included in the analysis in addition to the original observational series. The number of lags to be considered can be defined by estimating partial autocorrelation (see method in Supplementary material S5 in S1 File). In the present case, the biomass dataset was lagged by 1

year, and trophic flow dataset was lagged by 3 years. Original and lagged datasets were merged in single data tables (378000*14 for biomass data table and 375000*72 for trophic flow data table). PCA was performed on the new matrix using the ade4 package [39]. To compare the results of the NDND simulations against reconstructed food-web configurations, we projected the reconstructed data for the reference period 1986–2019 into the PCA space.

To identify the nature of trophic controls, we used correlation measures between species following the approach in Johannesen *et al.* [8]. Negative correlations and positive correlations are interpreted as top-down and bottom-up control, respectively. To identify persistent patterns of trophic control in the food-web, we then grouped biomass trajectories based on the similarity of their correlation patterns. Correlation patterns refer to the partial Pearson correlation matrix for each simulation. Partial correlation measures the direct correlation between two variables while accounting for the indirect correlation between the same two groups [40]. The correlation matrix is calculated from the lagged biomass data table (i.e. the data table containing the original observational series and the time-lagged series) and therefore includes instantaneous correlations but also lagged correlations (e.g. prey in years *t* with predator in year *t+1*, and vice versa). A complete linkage hierarchical clustering was performed on all simulations using the Euclidean distance between correlation matrices as the distance measure. The relevant number of clusters was set visually based on the dendrograms, and average correlation matrices were presented for each cluster. This analysis identified trophic controls over centennial time scales.

To explore the possibility of transient correlations at shorter time scales and to compare trophic control in simulated biomass trajectories with trophic control found based on observed data, we removed the last 19 simulated years from each time-series and derived a set of shorter time-series of 40 years each. A similar cluster analysis was performed on these shorter time-series.

Finally, to explore the variability of trophic control at decadal time scales, we performed a 15-year centered sliding window marginal correlation analysis on the short biomass time-series (40 years). The same analysis was conducted on the reconstructed data. The results of this analysis were then compared to the results of Lindstrøm *et al.* [24] and Johannesen *et al.* [8].

## Results

### A. Simulation outputs

Simulated time-series of individual species displayed high variability within and between simulations (Fig 2). Life history and biological traits of individual species led to inter-specific variability of biomass dynamics (Fig 2).

Not surprisingly, the herbivorous and omnivorous zooplankton biomass displayed more rapid short-term and long-term changes than the other trophospecies: the simulated biomass of herbivorous and omnivorous zooplankton varied by two and three orders of magnitude, respectively. Benthos biomass varied by more than one order of magnitude. In the benthos group, the individual trajectories were more autocorrelated than for herbivorous and omnivorous zooplankton, and no long-term variability was observed. As omnivorous zooplankton, pelagic fish displayed biomass varying up to three orders of magnitude, but pelagic fish showed smaller short-term variability than omnivorous zooplankton leading to fewer shifts between configurations with higher pelagic fish biomass and configurations with lower pelagic fish biomass. Compared to previous trophospecies, demersal fish biomass displayed higher autocorrelation even though biomass could vary up to three orders of magnitude. The mammals biomass showed the lowest short-term and long-term variability of all trophospecies. Birds

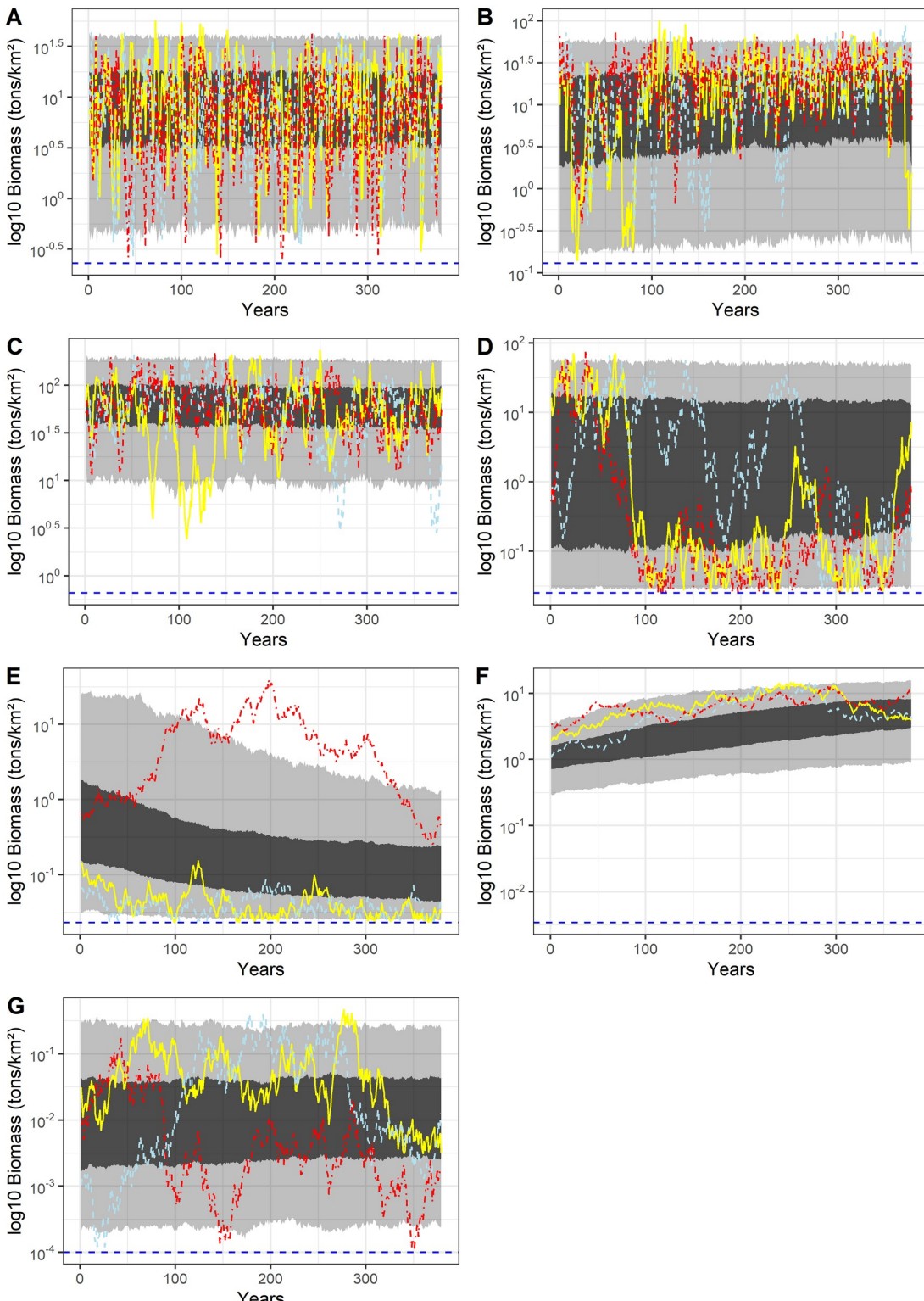

**Fig 2. Biomass time-series (log₁₀, tons.km⁻²) of Herbivorous zooplankton (A), omnivorous zooplankton (B), benthos (C), pelagic (D) and demersal fish (E), mammals (F) and birds (G).** The colored lines (red-dashed-detted, yellow-plain, and light blue-dashed) represent three randomly selected simulations. The dark blue dotted line represents the refuge biomass constraint. The light and dark grey areas contain 95% and 50% of all the simulated data, respectively.

biomass displayed higher variability on long-term scale with variations by approximately three orders of magnitude, but with limited year-to-year variations.

The CaN model reconstructs trajectories by including historical observations. Although observation data are not available for all the trophic groups specified in the model, the CaN model can provide a range of plausible biomass trajectories for each species, based on existing observations. As expected, the reconstructed data biomass envelopes of each trophospecies were smaller than those of simulated biomass and were included in the range of simulated biomass (Fig 3). For herbivorous and omnivorous zooplankton, benthos and birds, the

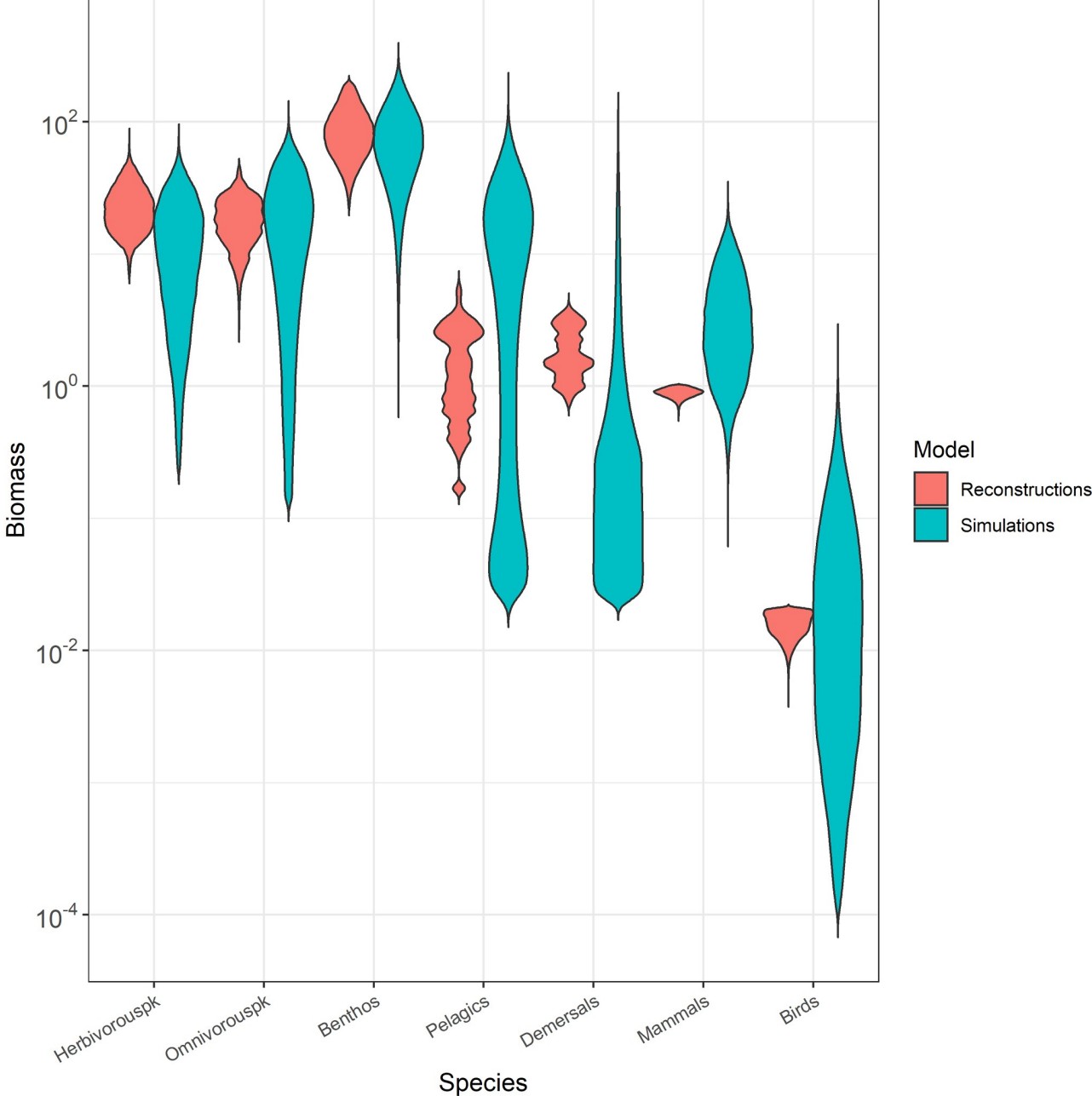

**Fig 3. Density distributions of biomass (tons.km$^{-2}$) in the NDND simulations (blue-right) and the CaN reconstructed biomass (red-left).** Biomass values are log10 transformed.

reconstructed data matched most of the simulated biomass as they overlapped with the densest part of the distributions. For pelagic and demersal fish and mammals, reconstructed data did not match the most frequent simulated biomass. The distribution of biomass for pelagic fish in the NDND model was bimodal with two "states" centered around high (~30 t.km$^{-2}$) and a low (~0.03 t.km$^{-2}$) biomass whereas the reconstructed data pointed toward intermediate levels (~1 t.km$^{-2}$). In the reconstructed data, demersal fish biomass were generally higher than most of the NDND simulated demersal biomass. On the contrary, the reconstructed mammal biomass was on average lower than the mammal biomass simulated with the NDND.

## B. Biomass configurations

The two first axes of the dPCA performed on simulated biomass explained 39.45% of variance (Fig 4). The projection of simulated (NDND) and reconstructed (CaN) biomass showed that reconstructed biomass overlapped with the simulation space (Fig 4). The reconstructed biomass configurations were included within the space of the simulated configurations, which was consistent with the biomass distributions from individual species (Fig 4). Biplot of the dPCA on simulated biomass highlighted two configurations of biomass. The first was aligned with the first axis of the dPCA and represented an opposition between pelagic fish, bird and omnivorous zooplankton biomass with pelagic fish and bird biomass being strongly positively correlated. The second was aligned with the second axis of the dPCA and reflected the opposition between marine mammals, demersal fish and benthos with benthos and demersal fish positively correlated. Interestingly, the reconstructed data displayed higher biomass of demersal fish and benthos, and low biomass of mammals and omnivorous zooplankton. In other words, past configurations of the Barents Sea food-web were dominated by demersal fish and benthos.

## C. Trophic pathways

The two first axis of the dPCA performed on the simulated flows explained 39.3% of variance (Fig 5). Complete overlap between simulated flows (NDND) and reconstructed flows (CaN) was observed (Fig 5A). Simulated flows formed three configurations of flows. The first configuration corresponded to the case where flows entering and exiting the pelagic fish group towards mammals and birds were high (Fig 5B). The second configuration was characterized by high flow rates between phytoplankton, omnivorous zooplankton, and mammals. The third configuration reflected to cases where flows entering and exiting the demersal fish group were high. In other words, the first main configuration corresponded to a pelagic trophic pathway, the second one described a short pathway from plankton to mammals and the third one reflected a benthic-demersal pathway. In the second configuration, both fish groups are completely bypassed by mammals feeding directly on omnivorous zooplankton. The density plots indicated that the pelagic and plankton-mammals pathway occurred more often than the benthic-demersal pathway in the simulated flows. Reconstructed flows were characterized by a benthic-demersal pathway as flows entering and exiting the demersal fish group were higher in the reconstructed data (Fig 5B). The pelagic fish-demersal fish flow was not included within any configuration. We assume that it is because the flow is part of the pelagic and the benthic-demersal pathways. In the benthic-demersal pathway, the pelagic-demersal flow is an income of biomass for the demersal fish, whereas in the pelagic pathway, it corresponds to removed biomass from the pelagic species.

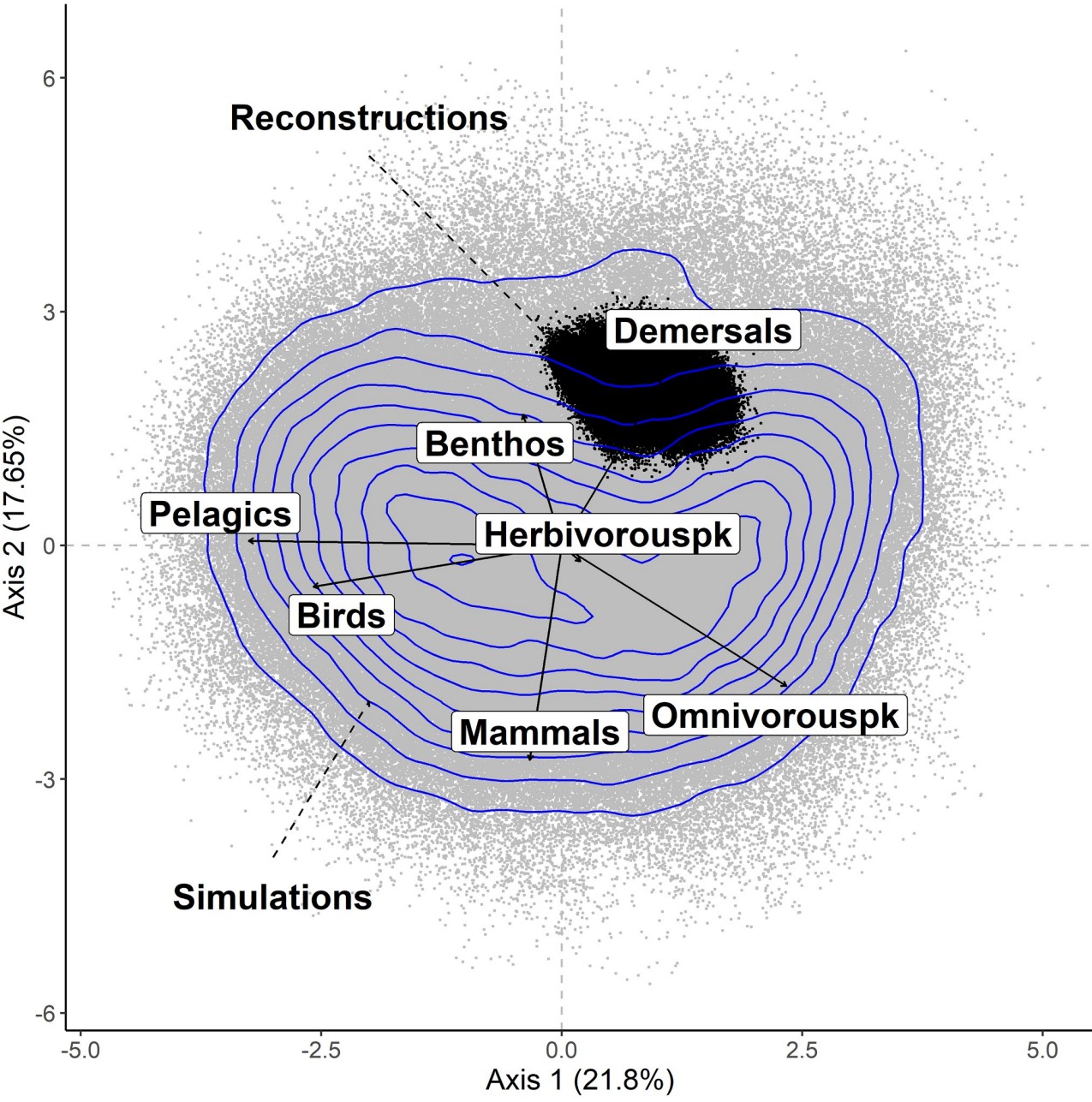

**Fig 4. Biplot of the dynamical principal component analysis (dPCA) performed on the simulated biomass time-series with the NDND, showing individual (grey points) and the projection of the original variables (arrows).** The projections of the reconstructed data configurations in the dPCA space are shown as black point. Blue lines show the contours of the density of observations in the simulated data only. The percentage of variance explained is reported on each axis.

### D. Trophic control

Distances between partial correlation matrices of long time-series (379 years) revealed small distances between the single partial correlation matrices indicating that partial correlation matrices of single time-series were similar. The maximum measured distance was slightly below 0.06 (see Supplementary materials S6 in S1 File). We cut the dendrogram in four

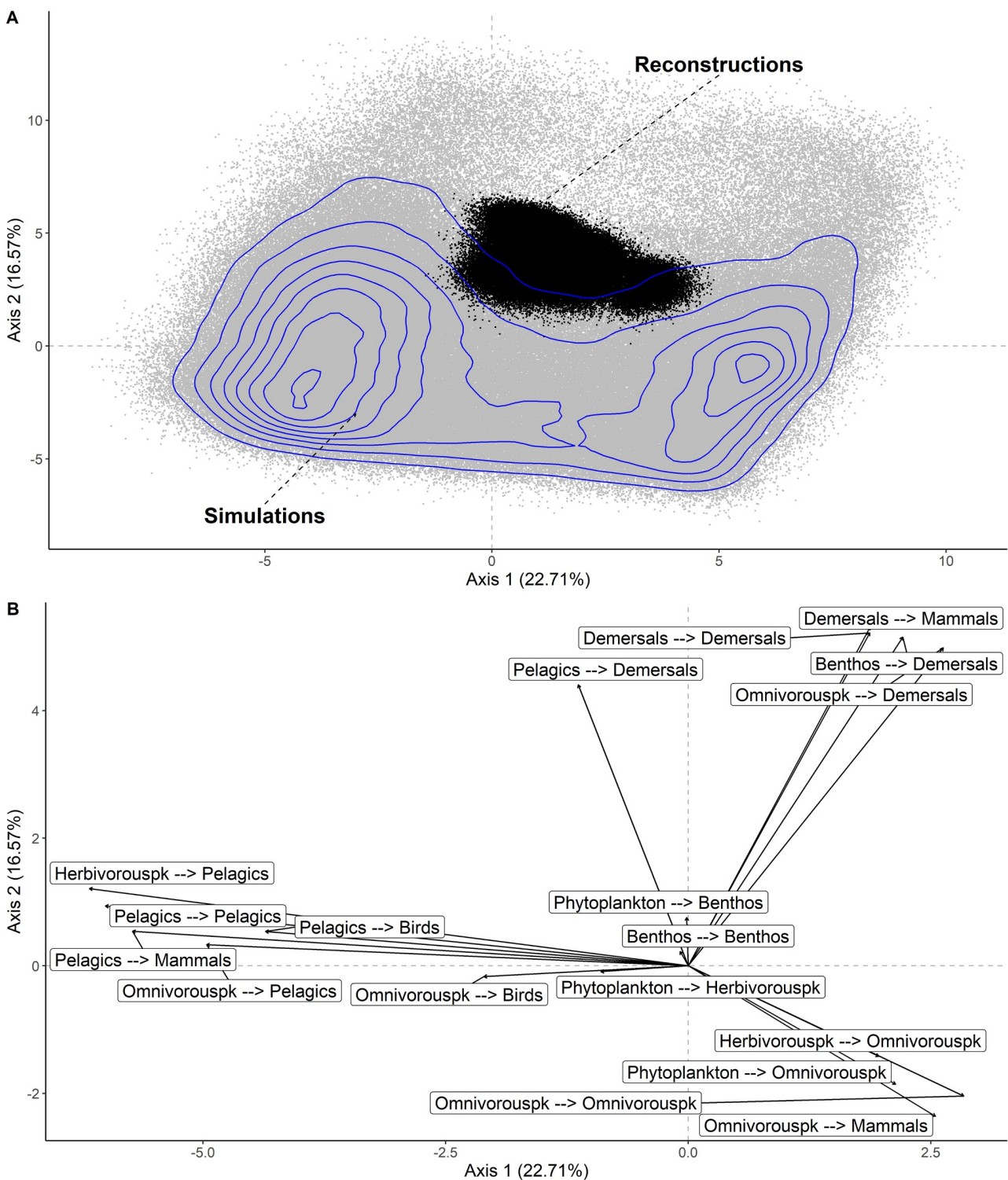

**Fig 5. Dynamical principal component analysis (dPCA) on fluxes generated by the NDND and CaN models and displayed as scatter plot (A) and projection of initial variables (B).** Each individual point represents a simulated flow configuration at time step t (grey points). The projection of reconstructed data is represented as black points. Blue lines show the contours of the density of observations in the simulated data only. The percentage of variance explained is reported on each axis.

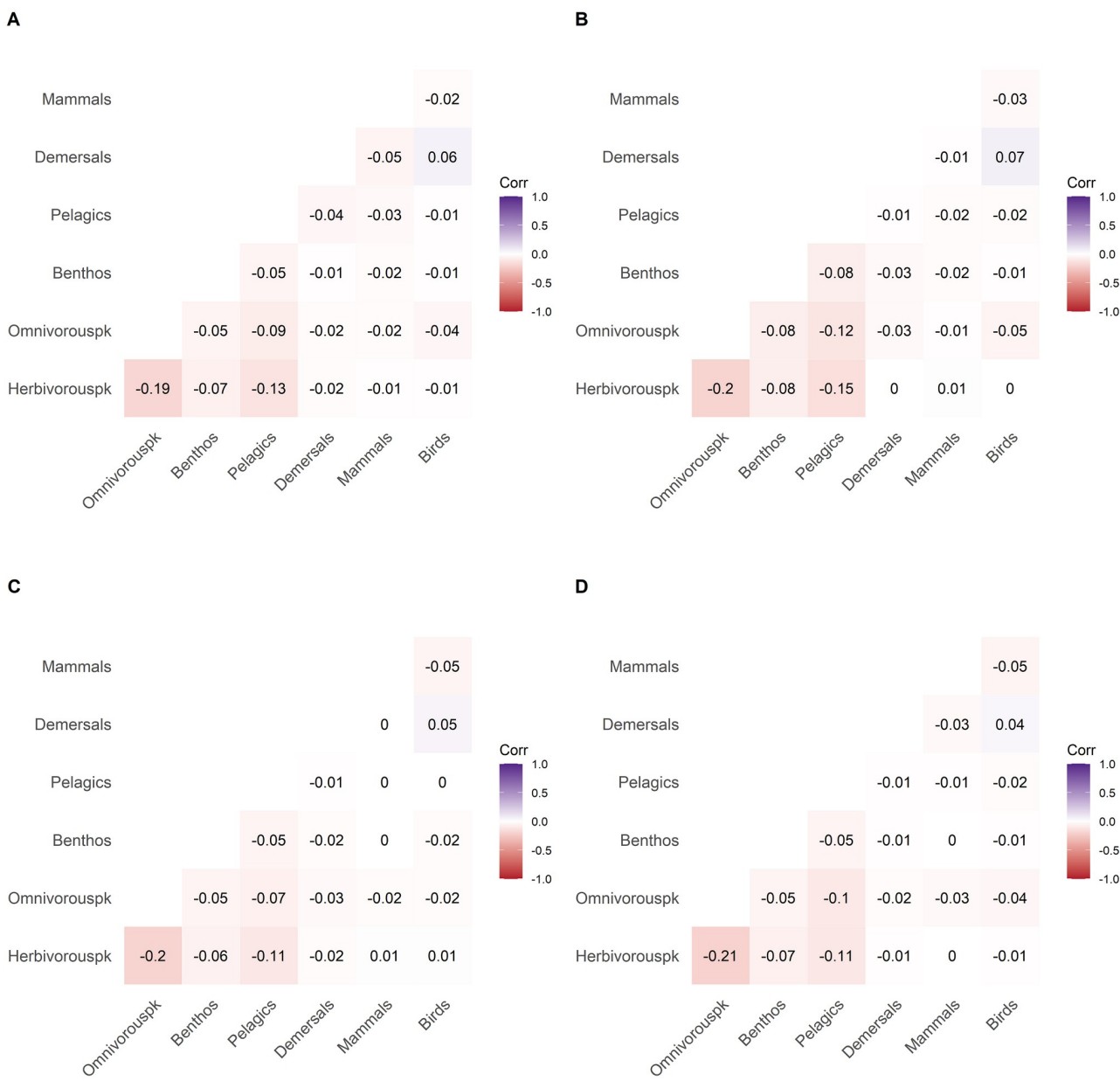

**Fig 6. Correlation plots of mean partial correlation matrices of clusters 1 (A), 2 (B), 3 (C) and 4 (D) defined in the hierarchical clustering.** Values in the plot correspond to mean Pearson partial correlation values between trophospecies. Red and blue colors indicate negative and positive correlations between two trophospecies, respectively. Negative and positive correlation represent bottom-up and top-down control, respectively.

clusters. The number of time-series in each cluster was uneven ($n_1 = 135$, $n_2 = 274$, $n_3 = 311$, and $n_4 = 280$).

The four clusters displayed few correlation patterns, i.e. the mean partial correlation of each cluster displayed no correlation pattern, and very little difference was identified between clusters (Fig 6). Most of the correlation values were between -0.21 and 0.07. Top-down control of herbivorous zooplankton by omnivorous zooplankton was present in all four clusters but correlation values were relatively low (between -0.21 and -0.19). Top-down control of pelagic fish on herbivorous zooplankton was found for all four clusters but again correlation values were

weak (-0.13, -0.15, -0.11 and -0.11 respectively) indicating weak trophic control. The absence of strong correlation values indicates that no long-term persistent trophic control can be highlighted.

The distances between 9000 partial correlation matrices of short biomass time-series (40 years) were higher than those between longer time-series. The maximum distance measured was 0.6 (see Supplementary materials S6 in S1 File). Thus, there were important differences between partial correlation matrices of single short time-series. We cut the dendrogram in three clusters. As for longer time-series, the number of time-series within the clusters was uneven ($n_1$ = 4786, $n_2$ = 3577 and $n_3$ = 627). Surprisingly, like long time-series (379 years), we observed almost no difference in correlation values between the clusters (Fig 7). Top-down control of herbivorous zooplankton by omnivorous zooplankton was found but the correlation values were relatively low (-0.21, -0.19 and -0.18 for each cluster respectively). Clusters 1 and 2 (Fig 7A & 7B) displayed higher correlation value between pelagic fish and herbivorous zooplankton than cluster 3 (-0.1 and -0.11 respectively). Cluster 3 (Fig 7C) was also characterized by negative correlation between mammals and benthos (-0.12). This interaction is indirect as no direct trophic link is defined between benthos and marine mammals. Furthermore, the correlation value was relatively low. Apart from the few correlation patterns mentioned above, no other significant correlation values were found indicating that for shorter time-periods (40 years) as for long time-series (379 years), no persistent trophic control was found.

The absence of correlation can result from two factors: (1) Either the correlation between two species stays close to 0, or (2) trophic control is variable within the same time-series, and higher and lower correlation values compensate each other leading to low correlation values. Estimating correlations on a sliding window, allows us to display the dynamics of trophic control within the same time-series. Correlation time-series displayed no trend (Fig 8A and 8C). 50% and 95% envelopes were centered around a correlation of 0. When time-series were considered separately, interdecadal shifts of trophic control were visible. Correlation between demersal and pelagic fish were found between -0.8 and 0.9 whereas correlations between pelagic fish and omnivorous zooplankton were found between -0.8 and 0.8 (Fig 8A and 8C).

Time-series of correlation between demersal and pelagic fish from reconstructed data ranged from -0.8 to 0.8 whereas correlations between pelagic fish and omnivorous zooplankton in the reconstructed data ranged from -0.75 to 0.5 (Fig 8B and 8D). Thus, reconstructed data were included in the NDND simulation range for demersal vs pelagic correlations. Reconstructed data also displayed variations of trophic control as observed in the NDND simulations, but the length of time-series was smaller for reconstructed data making difficult to observe possible cycles.

## Discussion

Studying the historical range of variability is essential to define a baseline for the variability of the Barents Sea ecosystem that can be used to design successful management policies [10]. However, past studies focusing on the variability of the Barents Sea food-web based on ecological time-series often define this baseline based on short time-periods representing only a fraction of the possible variability of the ecosystem. Thus, we explored the possible range of stochastic variability of the Barents Sea food-web using the NDND model. The primary aim of this study was to identify possible configurations of the Barents Sea food-web and to confront them to historical data. Our study shows that the range of possible food-web configurations is larger than the range observed during the last three decades.

Our analysis revealed four major configurations in the simulated biomass, which represent two types of opposition of biomass. The first opposed pelagic fish and birds to omnivorous

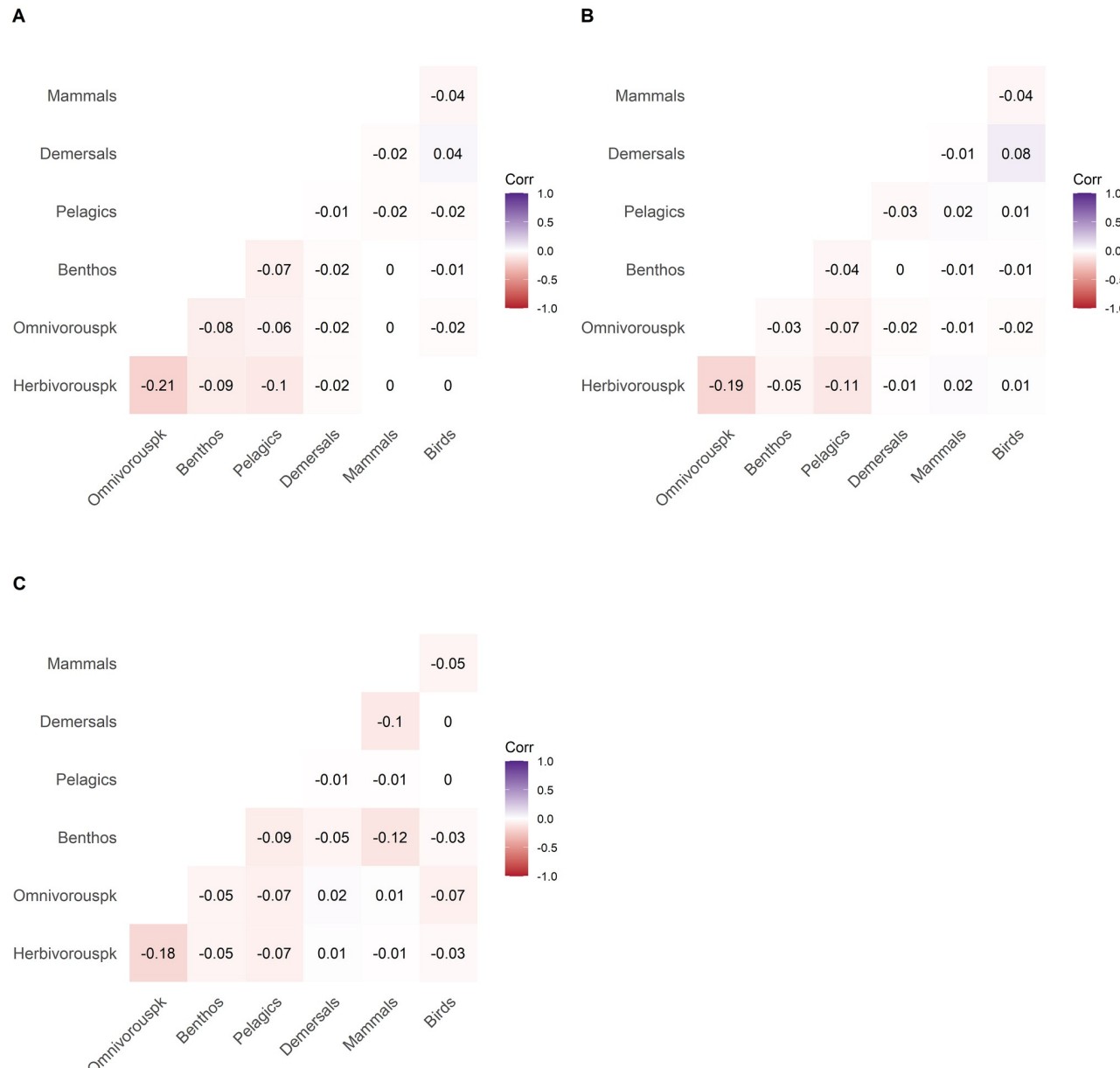

**Fig 7. Correlation plots of mean partial correlation matrices for clusters identified in the hierarchical clustering performed on short time-series (A-C for clusters 1–3 respectively).** Values in the plot correspond to mean Pearson partial correlation values between trophospecies. Red and blue colors indicate negative and positive correlations between two trophospecies, respectively. Negative and positive correlation represent bottom-up and top-down control, respectively.

zooplankton, whereas the second opposed demersal fish and benthos to marine mammals. The projection of reconstructed data in the dPCA space (Fig 4) indicated that the current state of the Barents Sea ecosystem corresponds to a configuration dominated by demersal fish and benthos. Reconstructed data displayed higher demersal fish biomass and lower mammals biomass than the majority of simulated biomass in the NDND simulations (Fig 3). These results are consistent with the hypothesized competition between mammals and demersal fish in the Barents Sea [41] and with the lower levels of estimated marine mammals abundance during the last three decades in the Barents Sea [42]. The simulated pelagic fish biomass displayed a

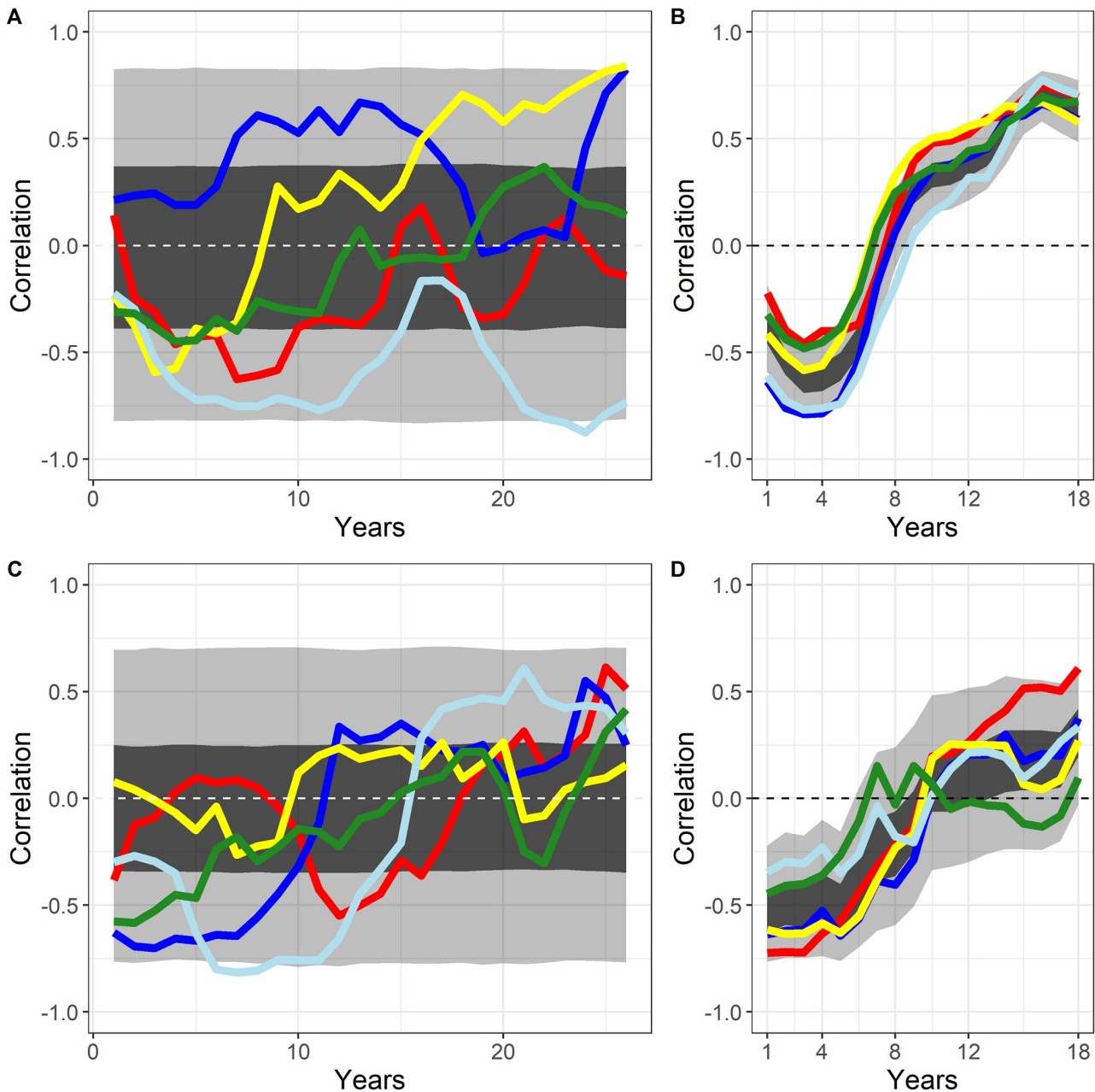

**Fig 8. Correlation estimated on a 15-year centered sliding window between demersal vs pelagic in the NDND simulations (A) and CaN simulations (B), and between pelagic vs. omnivorous zooplankton in the NDND simulations (C) and CaN simulations (D).** The white/black dotted line indicates when the correlation is null, the black envelope contains 50% of measured correlations, the grey envelope constraints 95% of all measured correlations. Colored lines correspond to correlation values of five randomly selected short time-series (40 years).

bimodal distribution with many high and low values of biomass, and few intermediate biomass values (Fig 3). Reconstructed data for pelagic fish biomass corresponded to the intermediate situation. A possible interpretation is that pelagic fish populations are in a transition phase between two states and could possibly substantially decrease or increase in the future, towards the levels observed in the simulations.

Past studies described three configurations of the Barents Sea food-web based on observed data for the last five decades [8,25], which are believed to have resulted from collapses of pelagic fish stocks, predation of pelagic fish by demersal fish, and management policies applied for pelagic and demersal fish groups. Our analysis pointed to four contrasted configurations that are partly consistent with the ones described in previous studies. Marine mammal-dominated configurations have not been reported in previous studies. In past dynamics of the Barents Sea, omnivorous zooplankton biomass increased when the pelagic fish biomass collapsed [25,27]. Our results are consistent with this observation as the configuration with low pelagic fish biomass was also characterized by high omnivorous zooplankton biomass. The pelagic fish-dominated configuration we define in this study is also characterized by a higher biomass of birds. This result is in line with previous studies showing that bird population size is linked to the biomass of pelagic fish in the Barents Sea, with observed declines in the population size of birds at times of pelagic fish stock collapses [43].

Our results show that there are three different trophic pathways of biomass in the NDND simulations. The pelagic and planktonic pathways appear to be the most common even though, our results suggest that the benthic-demersal pathway is also possible. Interestingly, the trophic pathway found in reconstructed data corresponded to the benthic-demersal trophic pathway. Trophic pathways in the simulated data emphasize the key role of pelagic fish for transferring energy from lower trophic levels to top-predators [27,44,45]. However, the pelagic fish species can be bypassed by the mammals feeding directly on omnivorous zooplankton in the NDND simulations. We assume that this pathway of energy corresponds to configurations of the food-web where biomass of pelagic fish is low and not sufficient to maintain the biomass of mammals. The benthic-demersal trophic pathway we identified is in line with trophic pathways identified using an Ecopath modelling approach [26]. Even though we found a complete overlap between simulated and reconstructed flows, reconstructed flows corresponded to configurations occurring less frequently than the pelagic and planktonic trophic pathways in the simulated flows. This indicates that according to the assumptions we have made, the dominant pathway of the Barents Sea food-web during the last three decades did not correspond to the dominant pathways identified in the simulations. An external factor, not accounted for in our study, might be forcing the Barents Sea food-web into the benthic-demersal pathway (i.e. the values of the trophic flows involving the demersal fish are higher in the reconstructions than in the simulations).

Previous studies exploring trophic relationships in the Barents Sea [46,47] have suggested that zooplankton species could be controlled by pelagic fish. However, a more recent study suggested bottom up-control of pelagic fish by zooplankton combined with top-down control of zooplankton by pelagic fish [48]. A study of demersal fish colonizing the northern part of the Barents Sea and the associated rapid decline in stomach fullness also suggested a top-down control [49]. In contrast to these studies, we found no persistent trophic control in the Barents Sea food-web neither over centennial (Fig 6) nor over decadal time scales (Fig 7). Rather, trophic controls tend to fluctuate on inter-decadal time scales and are highly variable between time-series (Fig 8). This variability of trophic control is found for most of the trophic interactions (see Supplementary material S7 in S1 File). Previously, in the Barents Sea food-web, trophic control shifted between top-down, bottom-up and wasp-waist processes [8,27]. The analysis of the NDND simulations suggests that this pattern of interannual variability in trophic control [8,24] is to be expected, even in situations in which trophic flows are varying at random.

The difference in the range of variability between our reconstructions and our simulations ensues from the differences in the constraints between the CaN model and the NDND model.

The CaN model and the NDND model are based on the same modelling principles: chance and necessity. Yet, the CaN model is more constrained than the NDND model since it integrates past observations to constrain the food-web trajectories [34]. Thus, we could expect the range of variability of the CaN reconstructed food-web configurations to be smaller than the range of variability of the NDND simulated food-web configurations.

The NDND model considers all configurations derived from a small number of biological and physical constraints. Hence, just as we argue that large deterministic models are too constrained and thus are not able to generate the natural variability of food-webs, it is fair to consider that the NDND simulations are under-determinate and that their variability could be greater than the real range of variability of the food-web.

In the NDND model, the dynamics of the Barents Sea food-web is driven by trophic flows [22,23]. At any time-step, the flows are drawn randomly (within the model constraints) and their values determine the species biomass at the next time step. This in turns modifies the constraints and thereby affects the drawing of the trophic flows at the following time step. The overlap between simulated and reconstructed food-web configurations is primarily driven by chance. It indicates that the reconstructed food-web configurations are part of a wider set of configurations that reflect the possible range of stochastic variability of Barents Sea food-web. Although exploring the range of historical variability is essential to define a baseline for the variability of marine ecosystems for management purposes [10], it is a challenge to measure it. In management policies, stochastic variability is described as a source of uncertainty in ecosystem models [50]. The model we use in this study explores the possible range of stochastic variability of the Barents Sea food-web instead of considering it as a source of uncertainty. Our findings can be useful to management as it provides a baseline for the variability of the Barents Sea food-web which can be used to assess the future changes in the dynamics of the ecosystem.

## Conclusion

This study shows that the diversity of possible biomass configurations and trophic pathways in the Barents Sea food-web extends beyond what has been observed during the last three decades. We found that reconstructed biomass configurations as well as reconstructed trophic pathways are strongly overlapping with our simulations though they only represent a subset of possible situations. Our simulations indicate that the Barents Sea food-web is dominated by pelagic and planktonic pathways. Our analyses suggest four major types of biomass configurations, characterized by opposite patterns in the abundance of pelagic fish and omnivorous zooplankton on one side and demersal fish and marine mammals on the other. We found no evidence for persistent trophic control in the Barents Sea food-web over centennial and multi-decadal scales throughout the food-web but revealed fluctuating top-down and bottom-up controls over interdecadal time scales.

## Supporting information

**S1 File. Revised supplementary information.**
(DOCX)

## Acknowledgments

We thank the reviewers for their helpful comments on this work. We would like to thank Bérengère Husson and Igor Eulaers their constructive feedback on the manuscript.

## Author Contributions

**Conceptualization:** Elliot Sivel, Benjamin Planque, Ulf Lindstrøm.

**Formal analysis:** Elliot Sivel, Nigel G. Yoccoz.

**Investigation:** Elliot Sivel, Benjamin Planque.

**Methodology:** Elliot Sivel, Benjamin Planque, Ulf Lindstrøm, Nigel G. Yoccoz.

**Supervision:** Benjamin Planque, Ulf Lindstrøm, Nigel G. Yoccoz.

**Visualization:** Elliot Sivel.

**Writing – original draft:** Elliot Sivel.

**Writing – review & editing:** Benjamin Planque, Ulf Lindstrøm, Nigel G. Yoccoz.

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
