## [Decision Letter · Decision Letter 0]

21 Jan 2021

PONE-D-20-39020

Multiple configurations and fluctuating trophic controls in the Barents Sea food-web

PLOS ONE

Dear Dr. Sivel,

Thank you for submitting your manuscript to PLOS ONE. After careful consideration, we feel that it has merit but does not fully meet PLOS ONE’s publication criteria as it currently stands. Therefore, we invite you to submit a revised version of the manuscript that addresses the points raised during the review process.

We look forward to receiving your revised manuscript.

Kind regards,

Charles William Martin

Academic Editor

PLOS ONE

Additional Editor Comments:

I have now received comment from 3 reviewers, in addition to reading through your submission myself. The reviews range from outright reject to accept. A major criticism brought up by reviewers is that your paper lacks appropriate context and is not written and set up as testing explicit hypotheses. Despite this, the methods appear solid and I am suggesting major revision is necessary for the manuscript to proceed. I urge the authors to seriously consider the reviewer comments (especially Reviewer 1) and restructure/rewrite the manuscript, especially the introduction and narrative structure to provide better rationale and clarity to the reader on the specific context and hypotheses that are being tested and dig a bit deeper into the driving mechanisms as suggested by Reviewer 3.

Reviewers' comments:

Reviewer's Responses to Questions

**Comments to the Author**

1. Is the manuscript technically sound, and do the data support the conclusions?

Reviewer #1: Yes

Reviewer #2: Yes

Reviewer #3: Yes

2. Has the statistical analysis been performed appropriately and rigorously? 

Reviewer #1: Yes

Reviewer #2: Yes

Reviewer #3: Yes

3. Have the authors made all data underlying the findings in their manuscript fully available?

Reviewer #1: Yes

Reviewer #2: No

Reviewer #3: Yes

4. Is the manuscript presented in an intelligible fashion and written in standard English?

Reviewer #1: Yes

Reviewer #2: Yes

Reviewer #3: Yes

5. Review Comments to the Author

Reviewer #1: General comments:: The rationale for the study is not strongly presented to emphasize the gap in knowledge or literature. There have been ample studies using multivariate approaches, that within a 50 year time period, show that the trophic interactions and food web structure are fluid and change over time. Lines 52-58 seem to be the place the authors are making their case for the study, but it is either not explicit enough or generally vague. If the authors are trying to address the uncertainty issue in ecological and food web modeling, it should be explicitly stated. Simply displaying if a food web can exhibit other configurations as referenced in Line 92 lacks the “teeth” of a robust scientific question. If this study was meant to explain and/or define the uncertainty in the modeled food web, then the interest in this study would increase exponentially. But that isn’t the case here as Line 166 states- ”A main goal of this study is to explore possible food-web configurations in the Barents Sea food-web and compare those to previously observed configurations.” While the statistical modeling approaches appear to be appropriate, it is the question, and thus rationale of the study that lacks. I would also disagree with Line 63 which states that food web models often do not have the ability to deal with internal variability. In fact, Ecopath with Ecosim, referenced in this study, for which there is a model created for the Barents Sea, has multiple ways to evaluate internal variability, the most notable being the Monte Carlo simulation plug-in, which has been widely applied and published. The Discussion also does not make the case for the analysis or why such a study would be important in future work or even what would be the next steps using their results as a jumping off point. The Discussion does not mention the limitations of the study, which should always be addressed when working with food web models.

Line 32: using an abbreviation not defined (RCaN)

Line 158: is the CaN the same as RCaN as used in the abstract. Should be consistent if the same model

Line 378: The fact that all pelagic species are lumped into a single functional group is a critical limitation of this study. Given that previous ecosystem configurations show high biomass of one pelagic species and a low biomass of the other (herring and capelin), lumping the groups together does not allow for a true examination of trophic dynamics. While herring and capelin may occupy a similar niche, the aggregation of the functional group detracts from interpreting what could be a mechanism of change in trophic configuration.

Line 401: The use of the word “Anyhow…” is not appropriate for scientific publication

Reviewer #2: Summary

The manuscript titled, “Multiple configurations and fluctuating trophic controls in the Barents Sea” uses a mass-balance ecosystem model to recreate various trophic structures and shifts in dominant species as seen in historical data. The authors use a well thought out methodological approach to simulate possible trophic controls, pathways, and structure for the Barents Sea, and compare it to historical data. They identified consistencies with their simulations and historical data in biomass configurations, but differences of possible trophic pathways and control between simulations and historical data. The number of simulated years was much greater than the number of years of historical observed data, thus it produced many more biomass configurations. The authors used correlation analyses to identify trophic pathways and potential controls. The methods are well described and the results well presented.

Overall, this manuscript is well structured and written, and is a valuable contribution to the scientific and modelling community. I endorse this article to be accepted for publication in PLoS ONE following minor revisions.

Major comments

Methods: I would actually like to see the NDND model equation in the main manuscript itself. It is an important component of the paper and understanding the model (and relatively concise).

Discussion: The authors should provide more discussion on the application of their model for possible future projections (i.e. climate change) and management potential (i.e. fishing regulations). For example, a mass-balance model such as Ecopath with Ecosim can be used with forcing functions and harvest control rules to simulate how multiple drivers and whether these drivers interact to affect ecosystem structure and dynamics.

Minor comments

L52-55: this sentence is a bit long and confusing. Please revise.

L123-124: missing word “…catch/stock to [be] equivalent…”

Reviewer #3: In this contribution, Sivel et al. theoretically explores if the Barents Sea food-web could be comprised of different food-web configurations. To this end, they conduct extensive simulations using a Non-deterministic Network Dynamic food-web model. The authors find that one of two food-web configurations is more similar to historical data than the other configuration. I find the study interesting and the chance by necessity modeling approach useful since it acknowledges the large uncertainties that comes with ecosystem modeling.

The major concern I have with this paper is that it does not discuss or explore different mechanisms that could give rise to the different food web configurations. Is it so that the harvest control rule imposed in the NTND model is responsible for the two emergent food-web configurations? I think that the authors should consider doing simulations with no fishing in order to explore if fishing could be the driving force behind the observed pattern. I think that these additional simulations could also help reveal why there are no trophic cascades observed for the full time series.

Here are some minor comments

I miss information on how fishing mortality was implemented in the CAN model!!

It would be good to complement figure 8 with lines showing “significant correlations”. The critical value for correlation coefficients for n=15 are for example ≈0.5.

Please check indexes of figure captions!

6. PLOS authors have the option to publish the peer review history of their article (what does this mean?). If published, this will include your full peer review and any attached files.

Reviewer #1: No

Reviewer #2: **Yes: **Travis C. Tai

Reviewer #3: No

---

## [Author Response · Author response to Decision Letter 0]

7 Mar 2021

Editor

A major criticism brought up by reviewers is that your paper lacks appropriate context and is not written and set up as testing explicit hypotheses. I urge the authors to seriously consider the reviewer comments (especially Reviewer 1) and restructure/rewrite the manuscript, especially the introduction and narrative structure to provide better rationale and clarity to the reader on the specific context and hypotheses that are being tested and dig a bit deeper into the driving mechanisms as suggested by Reviewer 3.

We considered the editor’s and reviewers’ suggestions and provided detailed answers to the comments of the three reviewers. We emphasized the rationale of this study, restructured the introduction, specified the hypothesis we had formulated for this study, and ran a supplementary simulation to examine the driving mechanisms as suggested by reviewer 3.

We want to clarify one point to avoid confusion while reading our answers. The rationale of this study relies on the lack of data for the Barents Sea ecosystem components for a time-period wider than five decades. However, the quality and quantity of the data significantly increased after 1985. Therefore, we reconstructed past configurations of the Barents Sea food-web for the time-period for which good data is available, i.e. the last three decades, and compared these reconstructions with the simulations of the possible Barents Sea food-web configurations.

Reviewer 1

The rationale for the study is not strongly presented to emphasize the gap in knowledge or literature.

We agree with reviewer 1 that the link between the rationale and the gap in knowledge, was not clearly presented in the original submission. The literature on the Barents Sea food-web provides an insight on its variability during the last five decades. However, these studies do not consider all trophic groups and thus, do not study the entire variability of the Barents Sea food-web. Furthermore, studies of the Barents Sea food-web variability for a time-period wider than the past five decades are not available in the literature.

To emphasize the gap in knowledge, the revised manuscript was modified as followed:

L 63-67: “As biomass data for marine mammals was lacking, these studies did not include marine mammals. A modelling study, however, estimated that the Barents Sea harp seal population experienced a strong decline in the 1950’s due to harvesting [15,16], but has partly recovered since. Higher biomass of marine mammals in the Barents Sea may result in alternate configurations that have not been observed yet.”

L 72: “However, this study [17] did not assess the variability of trophic pathways.”

L 73: “Trophic control in the Barents Sea food-webs remains unclear.”

We have also reorganized the structure of the introduction to emphasize that the studies of the variability of the Barents Sea food-web in the literature cover only 50 years, and that the tools used to study the variability of the food-web are not fit for purpose. Consequently, we present the NDND model as a tool suited to study the variability of the Barents Sea food-web.

There have been ample studies using multivariate approaches, that within a 50-year time period, show that the trophic interactions and food web structure are fluid and change over time.

We agree with reviewer 1 that several studies show fluctuations in food-web structure and trophic interactions over the last 50 years. An exhaustive list was provided by Möllmann and Diekmann (2012) for the North Pacific Ocean, the Eastern Scotian Shelf, the North Sea, the Baltic Sea, and the Black Sea. A shift from bottom-up to top-down control was documented for the Barents Sea ecosystem (Frank et al., 2007). However, for the Barents Sea, fluctuating trophic structure and trophic control over time was first suggested by Johannesen et al. (2012). Stige et al. (2019) provided a description of trophic control of pelagic fish on zooplankton in the Barents Sea as a global trophic control for the time-period 1980-2015. We describe the results from Johannesen et al. (2012) and Stige et al. (2019) in the revised manuscript as followed:

L 60-61: “Three different food-web configurations, reflecting successive shifts of dominance of capelin, herring and cod, have been identified in the Barents Sea [8].”

L 73-75: “Different trophic controls involving pelagic fish have been identified: a top-down control on mesozooplankton by pelagic fish [8], and a bottom-up control on capelin by krill [20].”

L 77-78: “Furthermore, gradual shifts in trophic control of the whole Barents Sea food-web have been identified in the last decades [8].”

Nevertheless, the data used by Johannesen et al. (2012) only covers the time-period 1970 to 2009 and Stige et al. (2019) does not describe the variability of trophic control in the Barents Sea food-web. In this article, we study the possible fluctuations in trophic structure and trophic control in a time-period wider than 1970-2015.

Lines 52-58 seem to be the place the authors are making their case for the study, but it is either not explicit enough or generally vague.

Following the editor and reviewers’ comments, we have restructured the introduction and link this paragraph to the studies on the variability of configurations of the Barents Sea food-web. This paragraph argues that the methods used to study the variability of the Barents Sea food-web are not appropriate because, either the length of the historical time series are too short, or the models used to simulate ecosystem dynamics are not able to reproduce the natural variability of the ecosystem. Therefore, we need an alternative method to study the variability of the Barents Sea food-web.

In the revised manuscript, L 79-83: “Ecological time-series used to describe the variability of marine ecosystems are often relatively short, typically less than 50 years, and the Barents Sea is no exception. Considering the limited amount of temporal data for some trophic groups (e.g. benthos, marine mammals, birds) and the lifespan of some species exceeding 50 years, it is difficult to make inference about future or past variability of the Barents Sea food-web without making assumptions that can result in large uncertainties [22,23].”

In the revised manuscript, L 88-90: “Despite this increase in complexity, these models are too constrained and not able to reproduce the temporal variability of the system [6,29].”

In the revised manuscript, L 90-91: “Thus, an alternative modelling approach is needed to study the variability of the Barents Sea food-web.”

If the authors are trying to address the uncertainty issue in ecological and food web modeling, it should be explicitly stated. Simply displaying if a food web can exhibit other configurations as referenced in Line 92 lacks the “teeth” of a robust scientific question.

We are not specifically addressing the ‘uncertainty’ issue as the reviewer suggests. Rather, we focus on the study of food web ‘variability’. We aim at comparing the possible variability of the Barents Sea food-web with its variability observed during the last decades. A major objective is to investigate if there are other possible configurations of the Barents Sea food-web and how they differ from those observed during the last three decades.

We clarified and rephrased the objective of this study in the revised manuscript: 

L 102-103: “In this study, we investigate the variability of the Barents Sea food-web configurations and how they differ from the configurations observed during the past three decades.”

If this study was meant to explain and/or define the uncertainty in the modeled food web, then the interest in this study would increase exponentially. But that isn’t the case here as Line 166 states- ”A main goal of this study is to explore possible food-web configurations in the Barents Sea food-web and compare those to previously observed configurations.”. While the statistical modeling approaches appear to be appropriate, it is the question, and thus rationale of the study that lacks.

We agree that defining uncertainties in the modeled food-web is interesting but, it is beyond the scope of this paper. The present study focuses on the variability of the Barents Sea food-web. Studying the variability of ecosystem is essential (1) to better evaluate the effect of external factors on the food-web and (2) to explore what changes can occur in the food-web without changes in the drivers. Whereas, the past literature focuses on the first point, this study investigates the second point.

I would also disagree with Line 63 which states that food web models often do not have the ability to deal with internal variability. In fact, Ecopath with Ecosim, referenced in this study, for which there is a model created for the Barents Sea, has multiple ways to evaluate internal variability, the most notable being the Monte Carlo simulation plug-in, which has been widely applied and published.

There might be a confusion between internal variability and uncertainty. With an Ecopath with Ecosim model, one can reproduce the past variability of an ecosystem (i.e. the changes in the past dynamics of the ecosystem) as the model outputs are fitted to historical time-series. However, the Monte-Carlo plug-in is used to capture “output variations due to base input parameter sensitivity” (Steenbeek et al., 2018). Heymans et al. (2016) used the Monte-Carlo plug-in as a tool to assess uncertainty and not internal variability of West Coast of Scotland food-web. Furthermore, Ecopath with Ecosim modelling relies on the “equilibrium” assumption which prevent the simulated dynamics to vary without external forcing when it is at equilibrium. Hence, with an Ecopath with Ecosim model, one can reproduce the past variability and assess the uncertainty of the model outputs, but cannot reproduce the internal variability of an ecosystem.

The Discussion also does not make the case for the analysis or why such a study would be important in future work or even what would be the next steps using their results as a jumping off point. The Discussion does not mention the limitations of the study, which should always be addressed when working with food web models.

We agree that the discussion does not make the case for the analysis. Therefore, we added a short section to the discussion in the revised manuscript:

L 359-363: “Studying the variability is essential to explore possible changes of the ecosystem which can occur in the future. However, past studies on the variability of the Barents Sea food-web lacks. Thus, we provide explored the variability of the Barents Sea food-web using the NDND model. The primary aim of this study was to identify possible configurations of the Barents Sea food-web and to confront them to historical data.”

In the revised manuscript, we present the relevance of the present study for future work and possible future developments as followed:

L 444-459: “In this study, we assessed the variability of the Barents Sea food-web resulting only from variation in trophic interactions, holding in particular temperature and impact of fisheries constant. Our findings can be considered as one baseline for the variability of the Barents Sea food-web. Harvesting of fish trophospecies has been explicitly included in the NDND model using HCRs. Some input parameters of the NDND model are temperature-dependent [31]. Hence, we can vary drivers such as fishing intensity and temperature to simulate possible dynamics of the Barents Sea food-web under different fishing and climate conditions. However, the NDND model simulates the largest range of possible trajectories of the food-web and we expect that the relevant forecast horizon [52] is quite short. However, Lindstrøm et al. [31] showed that ecological indicators estimated from NDND model outputs were compatible with ecological indicators estimated from historical data and other modelling studies. Hence, ecological indicators can be derived from the NDND model outputs to make projection of the possible future states of the food-web or to test management strategies on the harvestable stocks in the Barents Sea. The simplified food-web topology implemented in the NDND model does not allow to test single species management strategies. This can be achieved by increasing the specific resolution of the food-web topology, at the expense of a significant increase of the computational time."

Following the referee’s advice, we added a section on the limitations of the modelling approach used in the current study. We describe the limitations of this study and its implications in the revised manuscript.

L 436-443: “The NDND model considers all configurations derived from a small number of biological and physical constraints. Hence, just as we argue that large deterministic models are too constrained and thus are not able to generate the natural variability of food-webs, it is fair to consider that the NDND model is not constrained enough, because it relies on too few groups and processes. Hence, the NDND model could display a larger variability than the real variability of the food-web and result in “impossible” food-web configurations. Identifying such configurations and assessing biological or physical constraints that would prevent them in the NDND model, could help us understand some minimal sets of constraints associated with specific configurations.”

Line 32: using an abbreviation not defined (RCaN)

The undefined abbreviation was removed from the abstract. 

L 30-32: “We perform food-web simulations using the Non-Deterministic Network Dynamic model (NDND) for the Barents Sea, identify food-web configurations and compare those to historical reconstructions of food-web dynamics.”

Line 158: is the CaN the same as RCaN as used in the abstract. Should be consistent if the same model.

The CaN corresponds to the abbreviation for ‘chance and necessity’ as indicated L 170. RCaN correspond to the R package needed to run a chance and necessity model on R. “RCaN” was removed from the abstract and appears now first when the the RCaN package is mentioned.

L 152-153 : “The sampling algorithm used in this study is the Complex Polytope Gibbs Sampling algorithm (cpgs) available in the RCaN package [38].”

Other references to the “RCaN model” were also changed to the “CaN model” in the revised manuscript. 

Line 378: The fact that all pelagic species are lumped into a single functional group is a critical limitation of this study. Given that previous ecosystem configurations show high biomass of one pelagic species and a low biomass of the other (herring and capelin), lumping the groups together does not allow for a true examination of trophic dynamics. While herring and capelin may occupy a similar niche, the aggregation of the functional group detracts from interpreting what could be a mechanism of change in trophic configuration.

Any food-web model requires that some species be aggregated (Olivier and Planque 2017) and the level of aggregation is dependent on the question asked. The aim of this study is to investigate trophic interactions at the ecosystem level, rather than to resolve interspecific interactions between single species. Ulanowicz et al. (2014) showed that aggregation of species into functional groups had no major impact on overall food-web properties. Therefore, we have kept the simplified food-web topology in which capelin and herring are aggregated into a single functional group.

Line 401: The use of the word “Anyhow…” is not appropriate for scientific publication.

The sentence was modified in the revised manuscript.

L 410-412: “The higher levels of observed reconstructed demersal fish biomass constrain the reconstructed demersal fish trophic flows to be higher than those generally sampled by NDND model.”

Reviewer 2

Methods: I would actually like to see the NDND model equation in the main manuscript itself. It is an important component of the paper and understanding the model (and relatively concise).

The NDND model equation is an important component of the model. However, this paper does not claim to be a methodological paper and as such we decided not to overload the main manuscript with methods and an exhaustive description of the model. Most of the model technical details are available from earlier articles (Planque et al., 2014; Lindstrøm et al., 2017). In addition, we provide the mathematical equations in supplementary material S1. 

Discussion: The authors should provide more discussion on the application of their model for possible future projections (i.e. climate change) and management potential (i.e. fishing regulations). For example, a mass-balance model such as Ecopath with Ecosim can be used with forcing functions and harvest control rules to simulate how multiple drivers and whether these drivers interact to affect ecosystem structure and dynamics.

Following the reviewer 2 suggestion we added a paragraph on this topic in the discussion.

L 444-459: “In this study, we assessed the variability of the Barents Sea food-web resulting only from variation in trophic interactions, holding in particular temperature and impact of fisheries constant. Our findings can be considered as one baseline for the variability of the Barents Sea food-web. Harvesting of fish trophospecies has been explicitly included in the NDND model using HCRs. Some input parameters of the NDND model are temperature-dependent [31]. Hence, we can vary drivers such as fishing intensity and temperature to simulate possible dynamics of the Barents Sea food-web under different fishing and climate conditions. However, the NDND model simulates the largest range of possible trajectories of the food-web and we expect that the relevant forecast horizon [52] is quite short. However, Lindstrøm et al. [31] showed that ecological indicators estimated from NDND model outputs were compatible with ecological indicators estimated from historical data and other modelling studies. Hence, ecological indicators can be derived from the NDND model outputs to make projection of the possible future states of the food-web or to test management strategies on the harvestable stocks in the Barents Sea. The simplified food-web topology implemented in the NDND model does not allow to test single species management strategies. This can be achieved by increasing the specific resolution of the food-web topology, at the expense of a significant increase of the computational time."

L52-55: this sentence is a bit long and confusing. Please revise.

The sentence was corrected in the revised manuscript.

L 79-83: “Ecological time-series used to describe the variability of marine ecosystems are often relatively short, typically less than 50 years, and the Barents Sea is no exception. Considering the limited amount of temporal data for some trophic groups (e.g. benthos, marine mammals, birds) and the lifespan of some species exceeding 50 years, it is difficult to make inference about future or past variability of the Barents Sea food-web without making assumptions that can result in large uncertainties [22,23].”

L123-124: missing word “…catch/stock to [be] equivalent…”

The missing word was added (L 133-134: “[…] we assume the ratio catch/stock to be equivalent in biomass and numbers, […]”).

Reviewer 3

The major concern I have with this paper is that it does not discuss or explore different mechanisms that could give rise to the different food web configurations. Is it so that the harvest control rule imposed in the NTND model is responsible for the two emergent food-web configurations? I think that the authors should consider doing simulations with no fishing in order to explore if fishing could be the driving force behind the observed pattern. I think that these additional simulations could also help reveal why there are no trophic cascades observed for the full time series.

We agree that the question of the impact of fisheries on the food-web is valuable question. As suggested by the reviewer, we have performed simulations without fishing. Therefore, fishing mortality rate (Fmp in the main manuscript) was set to 0. We projected the biomass and flow configurations of the simulation without fishing in the dynamical principal component analysis (dPCA) space of the dPCA performed on the simulation with fishing to compare the spread of food-web configurations of each simulation in the dPCA space .

The projection of simulated biomass and flows shows a strong overlap between the configurations of the simulation with fishing and the configurations of the simulation without fishing (Fig. 1 and 2). It indicates that the food-web configurations of both simulations are very similar. We made the same observation for the projection of the variables in the dPCA space.

Fig 1. Dynamical principal component analysis (dPCA) performed on the simulated biomass time-series with the NDND. Blue points correspond to the simulation with fishing, red points correspond to the simulation without fishing and black points correspond to reconstructed biomass using CaN modelling. Projection of reconstructed biomass in the state space of the simulation with fishing (A) and without fishing (B). Comparison of distribution of simulated biomass in the dPCA space for simulations with fishing and without fishing (C). Projection of individual variables in the dPCA space (D).

Fig 2. Dynamical principal component analysis (dPCA) performed on the simulated flows time-series with the NDND. Blue points correspond to the simulation with fishing, red points correspond to the simulation without fishing and black points correspond to reconstructed flows using CaN modelling. Projection of reconstructed flows in the state space of the simulation with fishing (A) and without fishing (B). Comparison of distribution of simulated flows in the dPCA space for simulations with fishing and without fishing (C). Projection of individual variables in the dPCA space (D).

Fisheries in the NDND model are considered as fully compensatory (i.e. losses due to fishing is removed from the available biomass for predation) and not varying over time. However, the lack of response of the simulation output to the absence of fisheries is surprising and suggests an alternate assumption of the effect of fisheries on the Barents Sea food-web. Hansen et al. (2019) showed that fisheries may have an additive effect (i.e. the loss to fishing is added to the loss due to predation) on higher trophic levels of the Barents Sea food-web. However, the current NDND model is not appropriate to explore alternative assumptions of the effects of fishing on the food-web. Thus, it requires deeper investigations which will be presented in a future independent work. 

With the simulation without fishing, we bring new results to the analysis. However, these results are beyond the scope of the question we ask in this study. Therefore, we do not want to include them in the main manuscript or in the supplementary materials.

I miss information on how fishing mortality was implemented in the CAN model!!

The information on the implementation of fishing mortality was added in the manuscript.

L170-172: “Fishing in the CaN model is reconstructed as a non-trophic flow that is constrained by historical landing time-series for omnivorous zooplankton, pelagic fish, demersal fish, benthos, and marine mammals.” 

Time-series of landings included in the CaN model are presented in supplementary materials S4.

It would be good to complement figure 8 with lines showing “significant correlations”. The critical value for correlation coefficients for n=15 are for example ≈0.5.

We understand the comment from the reviewer. However, we do not aim at testing if the correlations are significant, but we focus on the pattern displayed in sliding correlations. In addition, the critical values for the correlation coefficients may be biased if the observations in the time series are autocorrelated (which is often the case). Therefore, we decided to not include the “significance lines” to the figure 8.

Additional comments:

After the first submission of the present study, we found some errors in the original code of the model. We corrected the code and re-ran the simulations used in this study. Parametrization of the model is identical. The data time-series used to constrain the CaN model were updated until 2019 (against 2013 in the original submission) and a new CaN simulation was also performed. 

The results with the new simulations are slightly different than those from the original submission, but general patterns and associated conclusions remain similar.

Figures 2 to 8 were updated according to the new results, as well as the material and method section, the result section, and the discussion section.

References

Frank KT, Petrie B, Shackell NL. The ups and downs of trophic control in continental shelf ecosystems. Trends in Ecology & Evolution. 2007;22: 236–242. doi:10.1016/j.tree.2007.03.002

Hansen C, Drinkwater KF, Jähkel A, Fulton EA, Gorton R, Skern-Mauritzen M. Sensitivity of the Norwegian and Barents Sea Atlantis end-to-end ecosystem model to parameter perturbations of key species. PLOS ONE. 2019;14: e0210419. doi:10.1371/journal.pone.0210419

Heymans JJ, Coll M, Link JS, Mackinson S, Steenbeek J, Walters C, et al. Best practice in Ecopath with Ecosim food-web models for ecosystem-based management. Ecological Modelling. 2016;331: 173–184. doi:10.1016/j.ecolmodel.2015.12.007

Johannesen E, Ingvaldsen RB, Bogstad B, Dalpadado P, Eriksen E, Gjøsæter H, et al. Changes in Barents Sea ecosystem state, 1970–2009: climate fluctuations, human impact, and trophic interactions. ICES Journal of Marine Science. 2012;69: 880–889. doi:10.1093/icesjms/fss046

Lindstrøm U, Planque B, Subbey S. Multiple Patterns of Food Web Dynamics Revealed by a Minimal Non-deterministic Model. Ecosystems. 2017;20: 163–182. doi:10.1007/s10021-016-0022-y

Möllmann C, Diekmann R. Marine Ecosystem Regime Shifts Induced by Climate and Overfishing. Advances in Ecological Research. Elsevier; 2012. pp. 303–347. doi:10.1016/B978-0-12-398315-2.00004-1

Olivier P, Planque B. Complexity and structural properties of food webs in the Barents Sea. Oikos. 2017;126: 1339–1346. doi:10.1111/oik.04138

Planque B, Lindstrøm U, Subbey S. Non-Deterministic Modelling of Food-Web Dynamics. Chiaradia A, editor. PLoS ONE. 2014;9: e108243. doi:10.1371/journal.pone.0108243

Steenbeek J, Corrales X, Platts M, Coll M. Ecosampler: A new approach to assessing parameter uncertainty in Ecopath with Ecosim. SoftwareX. 2018;7: 198–204. doi:10.1016/j.softx.2018.06.004

Stige LC, Eriksen E, Dalpadado P, Ono K. Direct and indirect effects of sea ice cover on major zooplankton groups and planktivorous fishes in the Barents Sea. ICES J Mar Sci. 2019;76: i24–i36. doi:10.1093/icesjms/fsz063

Ulanowicz RE, Holt RD, Barfield M. Limits on ecosystem trophic complexity: insights from ecological network analysis. Ecology Letters. 2014;17: 127–136. doi:10.1111/ele.12216

---

## [Decision Letter · Decision Letter 1]

15 Apr 2021

PONE-D-20-39020R1

Multiple configurations and fluctuating trophic controls in the Barents Sea food-web

PLOS ONE

Dear Dr. Sivel,

Thank you for submitting your manuscript to PLOS ONE. After careful consideration, we feel that it has merit but does not fully meet PLOS ONE’s publication criteria as it currently stands. Therefore, we invite you to submit a revised version of the manuscript that addresses the points raised during the review process.

We look forward to receiving your revised manuscript.

Kind regards,

Charles William Martin

Academic Editor

PLOS ONE

Journal Requirements:

Additional Editor Comments (if provided):

Two of the original three reviewers have commented on your revised manuscript. Unfortunately, their assessments indicate that your paper is still lacking in several aspects and comments have not been fully addressed. In fact, the original Reviewer 1 still suggests rejection with requests to alter the justification for the study, providing a scientifically justifiable reason for the modeling efforts and constructing a meaningful and useful model that can be used for hypothesis-driven questions. Importantly, they also suggest that the aggregation of pelagic species is inappropriate and I agree. Reviewer 3 again comments that the mechanisms are not well explored. I have suggested you be given a chance to again revise your manuscript, but I strongly implore you to incorporate these suggestions into the revised manuscript.

The manuscript could be greatly improved by considering the reviewer's comments, especially in the introduction and set up for the study. My suggestion is to consider what your model provides, why it is important/unique/novel, and how this as a tool could assist scientists and managers in answering questions moving forward. This would particularly be helpful in justifying your study and set it apart from numerous other modeling efforts.

Reviewers' comments:

Reviewer's Responses to Questions

**Comments to the Author**

1. If the authors have adequately addressed your comments raised in a previous round of review and you feel that this manuscript is now acceptable for publication, you may indicate that here to bypass the “Comments to the Author” section, enter your conflict of interest statement in the “Confidential to Editor” section, and submit your "Accept" recommendation.

Reviewer #1: (No Response)

Reviewer #3: All comments have been addressed

2. Is the manuscript technically sound, and do the data support the conclusions?

Reviewer #1: Partly

Reviewer #3: Yes

3. Has the statistical analysis been performed appropriately and rigorously? 

Reviewer #1: Yes

Reviewer #3: Yes

4. Have the authors made all data underlying the findings in their manuscript fully available?

Reviewer #1: Yes

Reviewer #3: Yes

5. Is the manuscript presented in an intelligible fashion and written in standard English?

Reviewer #1: Yes

Reviewer #3: Yes

6. Review Comments to the Author

Reviewer #1: The authors took great care in addressing concerns from the original submission, however, this manuscript still lacks a compelling contribution to the literature. For instance, the paper says that the Barents Sea is a well studied ecosystem, but then the lack of data is the reason for the study. That rationale does not seem reasonable. It is clear that more data became available after 1985, but even considering that change in data, the previous comment about “well-studied,” is confounded. In addition, every food web model is considered “one snapshot” of that particular food web, and it is well understood that there are many more possible configurations of a food web from the same system. So, the goal of the manuscript to show how many different configurations of one food web exist, does not provide a meaningful contribution to the system or to food web modeling in general. What would be meaningful, is to develop one food web model, using the best data available, and then asking scientific questions about the system using the model as a tool to answer those questions. In other words, the significance of this contribution still remains questionable. Another remaining concern that the authors chose not to address is the combination of pelagic species into one functional group. While many if not all food web models combine species into functional groups, if there are specific aspects and ecological interactions for which the authors are going to consider in the Discussion, then those species should be disaggregated so their trophic responses could be interpreted separately. The response of “resolving trophic interactions at the ecosystem level,” does not provide a substantial rebuttal to the suggestion of disaggregating species for which the authors make individual comments in the Discussion. A trophic interaction, by definition is at the group level, so “ecosystem level trophic interactions,” does not make much sense, unless the authors consider Information Theory and Network Analyses--which provide metrics for ecosystem structure and function.

Reviewer #3: I appreciate that the authors conducted additional analyses investigating the role of fishing for their result. I´m convinced that fishing was not a driving force of the gained result. However, I still lack a reasoning as to why one of the simulated biomass configurations is more similar to reconstructed food web configuration. I think the analyses are well done, but I still think that it would be good if the authors delved a bit deeper into the mechanisms leading to the simulated results before publication.

7. PLOS authors have the option to publish the peer review history of their article (what does this mean?). If published, this will include your full peer review and any attached files.

Reviewer #1: No

Reviewer #3: No

---

## [Author Response · Author response to Decision Letter 1]

28 May 2021

Editor: 

In fact, the original Reviewer 1 still suggests rejection with requests to alter the justification for the study, providing a scientifically justifiable reason for the modeling efforts and constructing a meaningful and useful model that can be used for hypothesis-driven questions.

We understand the concerns raised by reviewer #1. We have revised the introduction to further clarify the purpose of our study, explain why the modelling approach is suitable to achieve this purpose and why the results are of potential interest for the scientific community. We disagree with reviewer #1 about the significance of our research but we would like to remind the editor and reviewer that significance should not be considered since it is not part of the evaluation criteria retained for publication in PlosOne.

Importantly, they also suggest that the aggregation of pelagic species is inappropriate, and I agree.

The aggregation or splitting of species group should be justified by the model purpose. In the present case, we argue that disaggregating pelagic species would not constitute an improvement to address this purpose and that splitting the pelagic fish group will render the model more complex, more difficult to parametrize and the results more difficult to interpret. We present the detailed rationale for not splitting the pelagic trophospecies in smaller groups in the response to reviewer #1.

Reviewer 3 again comments that the mechanisms are not well explored. 

The NDND model presented in this study is not designed to explore mechanisms, but to simulate possible food-web dynamics given stochastic trophic flows that lead to biomass changes. The RCaN model is a NDND model constrained by data and thus it has a smaller range of food-web configurations. We argue that some simulated dynamics are similar to the reconstructed dynamics “by chance”. We clarify the mechanism of the NDND and the RCaN models in the response to reviewer #3 as well as in the material and methods section of the revised manuscript.

The manuscript could be greatly improved by considering the reviewer's comments, especially in the introduction and set up for the study. My suggestion is to consider what your model provides, why it is important/unique/novel, and how this as a tool could assist scientists and managers in answering questions moving forward. This would particularly be helpful in justifying your study and set it apart from numerous other modeling efforts.

We have revised the introduction section to emphasize the use of the range of variability of the Barents Sea food-web configurations to define a baseline for the ecosystem variability for management purposes (L 51-53 in the revised manuscript: “The historical range of variability informs us on how an ecosystem has varied in the past. It can be used as a reference for the variability of an ecosystem to improve the assessment and the prediction of future changes in its dynamics [10].” ; L 91-93 in the revised manuscript: “However, there is a need to define a baseline for the variability of the Barents Sea ecosystem to assess if the recent changes in its dynamics may reflect its stochastic variability or if they were induced by variations in anthropogenic drivers”). As it simulates the possible range of food-web configurations, the NDND model is a suitable tool to assess this baseline.

Reviewer #1: 

however, this manuscript still lacks a compelling contribution to the literature.

For instance, the paper says that the Barents Sea is a well studied ecosystem, but then the lack of data is the reason for the study. That rationale does not seem reasonable. It is clear that more data became available after 1985, but even considering that change in data, the previous comment about “well-studied,” is confounded.

The Barents Sea is a well-studied ecosystem compared to many other marine ecosystems. There is a large amount of data available for many species in the Barents Sea, for the last 3-5 decades. This is particularly true for fish and plankton species. Most monitoring programs in the Barents Sea started in the 1980’s. Except for the major commercial species such as Atlantic cod and herring, little is known about dynamics of other components before 1980. Even today, some components of the Barents Sea are poorly sampled and monitored, for example benthos. So, even though the Barents Sea is relatively well studied, there are many groups for which data is limited.

To avoid further confusion, we have removed the term “well-studied” from the revised manuscript.

In addition, every food web model is considered “one snapshot” of that particular food web, and it is well understood that there are many more possible configurations of a food web from the same system. So, the goal of the manuscript to show how many different configurations of one food web exist, does not provide a meaningful contribution to the system or to food web modeling in general. What would be meaningful, is to develop one food web model, using the best data available, and then asking scientific questions about the system using the model as a tool to answer those questions. In other words, the significance of this contribution still remains questionable. 

Although we understand the reviewer’s point, we disagree with it. Reviewer #1 points that “it is well understood that there are many more possible configurations of a food web from the same system”. Even if this is true, this does not mean that we know what these other configurations can look like or that we know the range of possible configurations. What we provide in this contribution, using a novel simulation approach, is a quantitative description of possible food-web configurations, including those that have not been observed in the recent past. This is a significant step forward. The results can have direct implications for the design of management policies that need to deal with uncertain future changes.

L 51-53 in the revised manuscript: “The historical range of variability informs us on how an ecosystem has varied in the past. It can be used as a reference for the variability of an ecosystem to improve the assessment and the prediction of future changes in its dynamics [10].” 

L 91-93 in the revised manuscript: “However, there is a need to define a baseline for the variability of the Barents Sea ecosystem to assess if the recent changes in its dynamics may reflect its stochastic variability or if they were induced by variations in anthropogenic drivers.”

The reviewer further suggests that it would be meaningful to “develop one food web model, using the best data available, and then asking scientific questions about the system using the model as a tool to answer those questions”. We have done just that. The model is based on decades of data collected in the Barents Sea and elsewhere to structure the food web and set the input parameter values. This model, which is a dynamic-stochastic food web model, is an appropriate tool to answer our question: what could be the range of configurations of the Barents food-web.

L 76-78 in the revised manuscript: “As such, the NDND model has been designed to reproduce the high variability of natural systems by exploring their “state-space”.”

Another remaining concern that the authors chose not to address is the combination of pelagic species into one functional group. While many if not all food web models combine species into functional groups, if there are specific aspects and ecological interactions for which the authors are going to consider in the Discussion, then those species should be disaggregated so their trophic responses could be interpreted separately. The response of “resolving trophic interactions at the ecosystem level,” does not provide a substantial rebuttal to the suggestion of disaggregating species for which the authors make individual comments in the Discussion. A trophic interaction, by definition is at the group level, so “ecosystem level trophic interactions,” does not make much sense, unless the authors consider Information Theory and Network Analyses--which provide metrics for ecosystem structure and function.

One should consider splitting functional groups in smaller groups if the aim of the study is to resolve interactions between specific sub-groups or species and the rest of the ecosystem. In other words, if the question of the study does not focus specifically on the trophic interaction between capelin and herring (and the interactions between these two species with other components of the food web), we don’t see the need to further complexify the food-web topology with respect to this specific group. Splitting the pelagic group in smaller groups is not more justified than splitting the demersal fish, benthos, or plankton groups, which are also heterogeneous. By increasing the complexity of the food-web, we might move away from the first purpose of the NDND model, which is to explore the stochastic variability of an ecosystem using a simplified representation of its real food-web.

We have revised the discussion section in the revised manuscript to emphasize the importance of trophic groups in discussion of Barents Sea food webs and removed any reference to species specific interactions. 

Reviewer #3: 

I appreciate that the authors conducted additional analyses investigating the role of fishing for their result. I´m convinced that fishing was not a driving force of the gained result.

However, I still lack a reasoning as to why one of the simulated biomass configurations is more similar to reconstructed food web configuration. I think the analyses are well done, but I still think that it would be good if the authors delved a bit deeper into the mechanisms leading to the simulated results before publication.

We understand the reviewer’s comment. The NDND model is made of a stochastic and a deterministic part. While the trophic flows are drawn randomly (given that they fulfil the model constraints), the subsequent changes in biomass are fully determined by the model master equation. In return, these new biomasses will set new constraints on the trophic flows in the next time step, and so on. We have edited the material and methods section to emphasize the mechanisms of the NDND model:

L 104-109 in the revised manuscript: “In the NDND model, the dynamics of the different trophospecies (hereafter named species) result from biomass exchanges between species whose values are sampled randomly (chance), given a set of biological and physical constraints (necessity). Trophic flows define mechanistically the biomass at the next time-step according to the master equation of the model (see supplementary materials S1). Estimated biomass values will then constrain the values of the trophic flows for the next time-step, and so on.”

The RCaN model used for reconstructing past dynamics relies on the same principles as the NDND model. In addition, RCaN integrates historical data about the species and fluxes to further constrain the dynamics of the food-web. In other words, RCaN is a NDND model constrained by historical data. NDND simulations explore a wider range of possible dynamics and some of these dynamics are similar to the RCaN dynamics “by chance”. We discussed this in two paragraphs in the discussion section in the revised manuscript:

L 420-426 in the revised manuscript: “The difference in the range of variability between our reconstructions and our simulations ensues from the differences in the constraints between the RCaN model and the NDND model. The RCaN model and the NDND model are based on the same modelling principles: chance and necessity. Yet, the RCaN model is more constrained than the NDND model since it integrates past observations to constrain the food-web trajectories [34]. Thus, we could expect the range of variability of the RCaN reconstructed food-web configurations to be smaller than the range of variability of the NDND simulated food-web configurations.”

L432 – 438 in the revised manuscript: “In the NDND model, the dynamics of the Barents Sea food-web is driven by trophic flows [22,23]. At any time-step, the flows are drawn randomly (within the model constraints) and their value determines the species biomass at the next time step. This in turns modifies the constraints and thereby affects the drawing of the trophic flows at the following time step. The overlap between simulated and reconstructed food-web configurations is primarily driven by chance. It indicates that the reconstructed food-web configurations are part of a wider set of configurations that reflect the possible range of stochastic variability of Barents Sea food-web.”

---

## [Editor Report · Decision Letter 2]

18 Jun 2021

Multiple configurations and fluctuating trophic controls in the Barents Sea food-web

PONE-D-20-39020R2

Dear Dr. Sivel,

We’re pleased to inform you that your manuscript has been judged scientifically suitable for publication and will be formally accepted for publication once it meets all outstanding technical requirements.

Kind regards,

Charles William Martin

Academic Editor

PLOS ONE

---

## [Editor Report · Acceptance letter]

29 Jun 2021

PONE-D-20-39020R2 

Multiple configurations and fluctuating trophic control in the Barents Sea food-web 

Dear Dr. Sivel:

I'm pleased to inform you that your manuscript has been deemed suitable for publication in PLOS ONE. Congratulations! Your manuscript is now with our production department. 

Kind regards, 

on behalf of

Dr. Charles William Martin 

Academic Editor

PLOS ONE